genetics, evolution, ecology

adaptive trait, endangered species, fitness estimation, genome-wide association study, indirect selection, wild population

**Authors for correspondence:**
Lukas Tietgen
e-mail: teitgen.lukas@gmail.com
Øystein Flagstad
e-mail: oystein.flagstad@nina.no

†Shared senior authors.
Special Feature: Wild quantitative genomics: the genomic basis of fitness variation in natural populations edited by Susan Johnston, Nancy Chen, Emily Josephs.

# Fur colour in the Arctic fox: genetic architecture and consequences for fitness

Lukas Tietgen[1,2], Ingerid J. Hagen[1,2], Oddmund Kleven[2], Cecilia Di Bernardi[2,3], Thomas Kvalnes[1], Karin Norén[4], Malin Hasselgren[4], Johan Fredrik Wallén[4,5], Anders Angerbjörn[4], Arild Landa[2], Nina E. Eide[2], Øystein Flagstad[2,†] and Henrik Jensen[1,†]

[1]Centre for Biodiversity Dynamics (CBD), Department of Biology, Norwegian University of Science and Technology (NTNU), Trondheim 7491, Norway
[2]Norwegian Institute for Nature Research (NINA), Trondheim 7485, Norway
[3]Department of Biology and Biotechnologies 'Charles Darwin', University of Rome La Sapienza, Viale dell' Università 32, Rome 00185, Italy
[4]Department of Zoology, Stockholm University, Stockholm 10691, Sweden
[5]Swedish Museum of Natural History, Stockholm 10405, Sweden

LT, 0000-0002-7479-6166; IJH, 0000-0003-1028-3940; OK, 0000-0003-0267-6795; CDB, 0000-0002-1171-1516; TK, 0000-0002-3088-7891; MH, 0000-0002-4875-4413; AL, 0000-0002-2533-5179; NEE, 0000-0002-7645-3300; ØF, 0000-0002-5534-8069; HJ, 0000-0001-7804-1564

Genome-wide association studies provide good opportunities for studying the genetic basis of adaptive traits in wild populations. Yet, previous studies often failed to identify major effect genes. In this study, we used high-density single nucleotide polymorphism and individual fitness data from a wild non-model species. Using a whole-genome approach, we identified the *MC1R* gene as the sole causal gene underlying Arctic fox *Vulpes lagopus* fur colour. Further, we showed the adaptive importance of fur colour genotypes through measures of fitness that link ecological and evolutionary processes. We found a tendency for blue foxes that are heterozygous at the fur colour locus to have higher fitness than homozygous white foxes. The effect of genotype on fitness was independent of winter duration but varied with prey availability, with the strongest effect in years of increasing rodent populations. *MC1R* is located in a genomic region with high gene density, and we discuss the potential for indirect selection through linkage and pleiotropy. Our study shows that whole-genome analyses can be successfully applied to wild species and identify major effect genes underlying adaptive traits. Furthermore, we show how this approach can be used to identify knowledge gaps in our understanding of interactions between ecology and evolution.

## 1. Introduction

Phenotypic variation that causes individual differences in survival or reproductive success may lead to adaptive evolution by natural selection [1]. Recent advances in molecular analytical methodologies and the increased availability of genomic data allow us to connect phenotypic variation in traits to their causal genes [2–4]. This enables us to directly assess the fitness consequences of genotypic variation and improve our understanding of the eco-evolutionary dynamics in wild populations.

A commonly used method for mapping genes for phenotypic traits is to conduct a genome-wide association study (GWAS) [5]. While being widely used to map genes of human diseases [6], the use of GWAS in wild animal populations is still somewhat limited [7,8]. Aside from methodological issues (e.g. sample size, density of genetic markers, relatedness and reproducibility of associations [7–9]), most studies that map genes underlying fitness-related traits find that these traits are polygenic and thus struggle to detect significant associations

between single genetic markers and the trait in question [10–14]. Nevertheless, some studies have shown that GWAS is capable of identifying single genes or genomic regions underlying fitness-related traits in wild populations. Johnston *et al.* [15] found the gene underlying polymorphism for horn morphology, an important fitness-related trait, in wild Soay sheep *Ovis aries*. Likewise, Barson *et al.* [16] discovered a large effect locus explaining variation in age at maturity, a highly variable and fitness-related trait in Atlantic salmon *Salmo salar*. Recently, another study found that also loci on other chromosomes explain some of the phenotypic variances in maturation time in Atlantic salmon, thus showing a polygenic basis nonetheless [17]. The adaptive significance of beak morphology in the different Darwin's ground finches *Geospiza* is well known and a GWAS was used to document a major effect region on chromosome 1A [18].

Coloration is one of the most conspicuous phenotypic traits in animals and has been the subject of research for decades, if not centuries [19]. Animal coloration can have many different purposes (e.g. camouflage, communication) [20] and effects of coloration on fitness have been shown in a wide range of animal species [21–24]. Because coloration is such a conspicuous trait, it is appealing to solely account differences in fitness to the colour phenotype. However, it is important to keep in mind that there might be more to a trait than the phenotype itself. Hadfield *et al.* [25] even showed that colour phenotypes do not always coincide with genetic patterns. Additionally, an association between coloration and other phenotypic traits, such as sexual behaviour, aggressiveness, stress response and energy homeostasis, has been shown in different species, suggesting pleiotropic effects of coloration genes [26,27]. Such covariation raises the question of how well we can predict evolutionary consequences of selection on a phenotypic trait when the genes underlying the trait are strongly linked to or affect (through pleiotropy) other phenotypic traits that themselves could affect fitness. Knowledge about causes of covariation between (potentially) fitness-related traits is, however, challenging to obtain for wild non-model species and demonstrates the importance of more studies aimed at gaining insight into the genetic architecture of adaptive traits.

The Arctic fox *Vulpes lagopus* is a species with interesting coloration features. It occurs in multiple distinct fur colour morphs and undergoes seasonal moult [28]. The two common colour morphs are described as the white and the blue morph without intermediate morphs. The third morph, called sandy, is extremely rare. White Arctic foxes have completely white winter fur, whereas their summer fur is mostly brown with lighter ventral sides. The blue morph is uniformly dark brown or charcoal year-round, with a slightly lighter coloration during winter. Fur colour in Arctic foxes appears to be inherited as a simple Mendelian trait with one autosomal locus, where the blue morph is a result of the effect of a dominant allele [29,30]. The white colour morph makes up over 90% of the global Arctic fox population [31]. Importantly though, the relative frequencies of the two morphs vary across the species distribution [32–35] and between different environments [34]. For example, in Iceland, the observed differences in colour morph frequencies are thought to reflect distinct selection advantages of the two colour morphs in different habitats [34]. The exact mechanisms underlying the global distribution of Arctic fox fur colour morphs are however not well studied or understood.

Previous molecular analysis suggested that two cysteine amino acid substitutions within the intragenic region of the melanocortin-1-receptor gene (*MC1R*) co-segregated with the Arctic fox fur colour morphs [29]. *MC1R* is known to regulate melanin-based coloration in a wide range of animal species [21,36,37], it is thus not surprising that *MC1R* may be involved in Arctic fox fur coloration. However, the study by Våge *et al.* [29] was designed as a candidate gene analysis and was not able to detect other genes possibly contributing to the colour morphs. With very few individuals and/or unknown genetic structures, the candidate gene approaches may also have various pitfalls [38]. *MC1R* is part of a gene family where five melanocortin receptors (*MC1R–MC5R*) share the same melanocortin ligands [26]. Pleiotropic covariation between melanin-based coloration and traits governed by *MC2R–MC5R* can thus be expected and is, in fact, found in different species [26].

For increased knowledge on the adaptive importance of variation in fur coloration, we assessed the genetic architecture of fur coloration and analysed fitness consequences of genetic variation related to this trait in a wild population of Arctic foxes. First, we used a whole-genome association analysis to examine the genetic basis and architecture of fur colour. Second, we quantified selection on fur colour genotypes using measures of fitness that link ecological and evolutionary processes. Finally, we investigated the potential for indirect phenotypic effects of fur colour genes through pleiotropy or physical linkage with other genes, and how these effects could affect the observed patterns of fitness and genotype frequencies.

## 2. Methods

### (a) Study species and data collection

In the early twentieth century, the Fennoscandian Arctic fox population was close to extinction. Despite protection since the late 1920s, the species did not recover, which led to the implementation of large-scale conservation actions across the Scandinavian peninsula, involving supplemental feeding, culling of red foxes *Vulpes vulpes* (the most important competitor of the Arctic fox) and a captive breeding programme [39,40]. The captive breeding programme is based on wild-born Arctic foxes held at a breeding station in Oppdal, Central Norway. Breeding pairs are chosen to represent all extant Scandinavian subpopulations to maintain genetic diversity. Arctic fox data used in this study originate from population monitoring [41] and the captive breeding programme, collected in the period 2007–2019. Complete life histories were available for foxes born at the breeding station and subsequently released ($n = 371$) and for foxes marked as pups during den surveys ($n = 810$; electronic supplementary material, table S3). For these foxes, DNA analyses, pit-tagging (RFID tags) and/or ear tagging were undertaken for later identification. Additionally, some foxes ($n = 206$) were only identified from scat sampling within the framework of the Norwegian National Arctic Fox Monitoring Programme [42,43].

The monitoring programme uses molecular tracking to document population trends annually and is also used to trace the establishment of Arctic foxes released from the captive breeding programme. Sampling of non-invasive material (faeces and hair) is carried out during winter and spring at known Arctic fox den sites across the species distribution [43]. During the study period, approximately 800 samples were collected and analysed each year. Individual identification was carried out by comparison of DNA profiles from samples that could be reliably genotyped to a database of known Arctic fox individuals,

including released foxes from the captive breeding programme, pit-tagged pups at the dens and non-invasively identified individuals from previous years.

## (b) Single nucleotide polymorphism genotyping, data quality control and genome-wide association study

From ear tissue DNA extracts, we successfully genotyped 701 Arctic fox individuals using a custom Affymetrix Axiom 702 k SNP array with 507 000 Arctic fox specific single nucleotide polymorphisms (SNPs). Only autosomal SNPs classified as poly high resolution [44] among our genotyped Arctic fox individuals were kept for the analyses (361 289 SNPs), and SNP positions were obtained from an Arctic fox reference genome assembly comprising 4048 scaffolds with SNP positions given within every scaffold [45]. See the electronic supplementary material, S2 for more details on the design of the array and quality control (QC) of the SNP data used herein. After QC, our genomic dataset consisted of 681 Arctic fox individuals (562 white, 119 blue) genotyped for 359 218 autosomal SNPs (electronic supplementary material, table S1).

A GWAS was used to investigate associations between autosomal genetic markers (SNPs) and the Arctic fox fur colour morphs. The analysis was performed using the *GenABEL* package in R [46] with fur colour as the response variable. A genomic relatedness matrix (GRM) was included in the model to account for relatedness. The *indep* function of PLINK was used with recommended parameters (50, 5, 2) to create a subset of 40 539 unlinked SNPs prior to the GRM calculation, to obtain most accurate relatedness estimates [47]. In the GWAS, a polygenic model including the full GRM was fitted, and a mixed model was used to test for association between Arctic fox fur colour and the genetic markers included in the study. Owing to genomic inflation ($\lambda = 1.92$), $p$-values were corrected for lambda (electronic supplementary material, figure S2a). To investigate whether additional correction for population structure was necessary, we reran the analysis including the first three principal components (PCs) achieved through classical multidimensional scaling. The genomic inflation factor ($\lambda = 1.902$) and the according quantile-quantile ($Q$–$Q$) plot (electronic supplementary material, figure S2b) were virtually unchanged. The three first PCs explained only *ca* 6, 5 and 3.5% of the total variation in the data. Removing the scaffolds with significant SNPs also removed the skew in the $Q$–$Q$ plot (electronic supplementary material, figure S2c, $\lambda = 0.91$), indicating that the unusually large number of highly significant SNPs could be generating the large skew and genomic inflation. Additionally, a cluster plot of the first two PCs did not reveal any structure concerning Arctic fox fur colour or origin (i.e. captive versus wild) (electronic supplementary material, figure S3). To account for multiple testing [5], we applied the Bonferroni correction, where the significance level ($\alpha = 0.05$) was divided by the number of SNPs included in the analysis [48].

To increase the number of individuals genotyped at the fur colour gene for selection analyses, we used a recently developed Fluidigm SNP array (I.J. Hagen, O. Kleven, L.G. Arntsen, J. von Seth, L. Dalen, N.E. Eide, Ø. Flagstad, H. Jensen 2018, unpublished data). This SNP array included 87 autosomal markers, including the SNP chosen to represent the Arctic fox fur colour genotype (AX-176934441; see Results). DNA genotyped on this platform was extracted from hair, scat and tissue. Nine hundred and twelve Arctic fox individuals were genotyped using the Fluidigm platform (electronic supplementary material, table S1). Of these, 109 were also genotyped using the Affymetrix SNP array. The AX-176934441 genotype was identical across the two SNP arrays in all these individuals. Of the remaining 803 individuals only genotyped using the Fluidigm platform, fur colour phenotype was known for 444 individuals (329 white, 115 blue). These individuals were used as a relatively independent dataset to verify the

association between the top GWAS SNP and fur colour because they were not included in the dataset used for the GWAS.

## (c) Selection analyses

Complete life-history data (annual survival and fecundity) were available for 1181 individuals from 2007 to 2018 in the Norwegian subpopulations (electronic supplementary material, figure S1 and table S3). Individuals were assigned to one of five age classes ($x = 1$–5). Thirty-five individuals older than 5 years were assigned to age class 5 to ensure sufficient sample size in each age class. Annual survival and fecundity were based on a range of sources: (i) observation and trapping during den surveys, (ii) DNA from faeces and hair samples, (iii) Biomark (Biomark, Inc., ID, USA) and Trovan Systems (Trovan Ltd, UK) RFID tag readers at feeding stations, and (iv) records from wildlife cameras. These sources allowed for a dataset with high resolution at an individual level. Arctic foxes suffer high mortality during winter (October–April) [49]. Thus, we used *pre-breeding census*, with each census covering the period from 1 April to 31 March the following year. The beginning of April coincides with the end of the mating season. Individual annual survival in census year $t$ was recorded as 1 for individuals that were inferred to be alive after 1 April in year $t + 1$ (otherwise 0).

Parentage was determined for 1497 individuals with known birth year and genotype, based on 85 autosomal SNPs, using the *Sequoia* R package [50] (electronic supplementary material, S4). The final pedigree was used to determine the number of pups that emerged from the den (and were genotyped) for each adult present in a subpopulation in a given year $t$. Annual fecundity was then determined as the number of pups that survived to recruit into the next year's population (i.e. were alive after 1 April next year $t + 1$). In addition, a dichotomous variable was made which was set to 1 if an individual had been found to breed in a given year $t$ (otherwise 0). Adults not recorded in the pedigree as parents of any pups in a given year $t$ were assumed not to have produced pups or bred that year. Because we used recruits as the base for the fecundity measure in this pre-breeding census framework, undetected pups (i.e. those that die quickly) do not affect the fecundity analysis. Despite extensive monitoring, it is expected that some observations are not recorded, given that the study population is a wild population spanning a large area. Still, recapture rates are high with only 10% of the study individuals being missed in one census year but reappearing later. The missing data are very likely random and not associated with our measurements. Thus, despite imperfect sampling, we do not expect systematic bias in our results.

### (i) Individual fitness

Selection on the fur colour genotype was estimated using a demographic model framework that uses reproductive value weighting to account for age structure and fluctuations in the age distribution [51–53]; see also the electronic supplementary material, S7. Using this framework, annual individual fitness ($\Lambda_i$) in a given year for individual $i$ in age class $x$ was defined as $\Lambda_i = W_i/v_x = (B_i v_1/2 + J_i v_{x+1})/v_x$ [54], where $W_i$ is the individual reproductive value, $B_i$ is the number of recruits produced, $J_i$ is the indicator of survival (1 if the individual survived, otherwise 0) and the $v$'s are age-specific reproductive values estimated from the mean projection matrices for males and females separately (electronic supplementary material, S7 and table S5). The reproductive value weighting ensures that $\Lambda_i$ is an age-independent measure of individual fitness, such that $E(\Lambda_i) = \lambda$, where $\lambda$ is the multiplicative growth rate of the population [52].

The relationships between fur colour genotype and annual individual fitness were modelled using generalized linear mixed effect models (GLMMs) with Poisson distribution, log

link function and random intercepts for *subpopulation* and *year*, fitted with the *lme4* package in R [55]. Models were fitted for females and males separately (see the electronic supplementary material, S7 for details). Likelihood ratio tests (LRTs) between models containing only the intercept and models containing *genotype* as predictor variable were performed to assess the effect of the genotype on fitness.

### (ii) Fitness components

To further investigate causes for any differences in fitness, the relationships between fur colour genotype and *fecundity* (i.e. number of recruits) and *adult annual survival* were analysed in separate models. Fecundity was modelled using a zero-inflated Poisson GLMM with log link function (electronic supplementary material, figure S7) and survival was modelled using a binomial GLMM and logit link function. In addition, the relationships between genotype and *breeding probability of adults* and the *recruitment probability of juveniles* (i.e. juvenile survival until at least 1 April the year following birth) were modelled using binomial GLMMs with logit link function. The analysis on juvenile recruitment probability was performed on a restricted dataset including only juveniles marked at the dens or released from the breeding station in order to be certain about their birth year ($n = 597$). As a starting point, all fitness component models included *genotype* and *sex* as fixed factors and a random intercept for *subpopulation*. Models with adult fitness components (fecundity, adult survival, breeding probability) included in addition *age* and *age*$^2$ as continuous covariates and random intercept for *year*, while models with recruitment probability included random intercepts for *birth year* and *den*. Interactions between *genotype* and *sex* or *age* were included to test whether the effect of genotype differed between sexes or changed with age. Statistical significance of the different variables was assessed using LRTs between models with and without the term of interest. In the case of non-significance, these terms were excluded (electronic supplementary material, table S6). The models were fitted using the *glmmTMB* R package [56] for zero-inflated models and the *lme4* R package for the remaining models [55].

### (iii) Environmental variables

Arctic foxes in this study were of two origins (wild- or captive-born). In the wild, the reproductive performance (number of litters and litter size) of the Arctic fox is to a large extent driven by food availability, varying strongly through the rodent cycle [57]. Although the Arctic fox is well adapted to winter severity and prey scarcity, the duration of snow cover could possibly explain geographical variation in the frequency of the two colour morphs [34]. Hence, for individual fitness and each of the fitness components, we tested whether the effect of colour genotype depended on *rodent phase*, duration of snow cover (i.e. *first snowfall* and *last snowfall*) or *origin* by fitting models with an interaction between an environmental variable and *genotype*, with separate models for each environmental variable (see the electronic supplementary material, S7 for further details). Models with individual fitness were fitted for females and males separately. Statistical significance was assessed using LRTs between models with and without the term of interest.

Heterozygosity advantage [58,59] could potentially be a reason for any differences in the fitness of fur colour genotypes. Hence, genome-wide heterozygosity was calculated for the 689 individuals genotyped on the Affymetrix platform using the *GenABEL* R package [46]. Differences in genome-wide heterozygosity were tested using a linear mixed-effects model with a Gaussian error distribution. Fur colour *genotype* and *origin* (i.e. captive- or wild-born) were included as fixed factor predictor variables. See the electronic supplementary material, S7 for further details.

To further investigate whether the observed differences in individual fitness coincide with the SNPs found to be significantly associated with Arctic fox fur colour, we performed a candidate region GWAS for individual fitness that included SNPs on Arctic fox scaffold 11 where significant SNPs were found in the fur colour GWAS (details in the electronic supplementary material, S9).

### (d) Gene analyses

BLAST searches [60], using BLAST+ 2.9.0 software [61], were performed to investigate genes located in the vicinity of SNPs that, based on the GWAS, were significantly associated with Arctic fox fur colour. An annotated Arctic fox genome is yet to be published, but there appears to be high synteny between dog and Arctic fox for large parts of their genomes [62]. Thus, the annotated dog genome *CanFam 3.1* [63] was used as the reference genome. See the electronic supplementary material, S3 for details.

Genes within 10 kb of significant SNPs were analysed for gene ontology (GO) term enrichment using the GOstat tool [64]. The distance of 10 kb was chosen to ensure strong linkage between the SNP and the gene. Owing to the lack of a dog-specific GO-database, the goa_human database was used. *p*-values for over-representation significance were corrected based on false discovery rate. Furthermore, for genes within 10 kb of a significant SNP that also was in high linkage disequilibrium (LD; $r^2 \geq 0.5$) with the top SNP, gene functions were investigated using the UniProt knowledgebase [65] and primary literature. These genes were also included in a GeneMANIA network analysis [66]. GeneMANIA uses a large dataset of functional association data to analyse relations and known co-expression between genes. GeneMANIA does not include a database for canines, thus the human database was used.

Unless otherwise stated, all analyses were performed in statistical software R v. 3.6.1 [67].

## 3. Results

### (a) Gene mapping

The GWAS revealed a total of 495 SNPs significantly associated with Arctic fox fur colour at a Bonferroni-adjusted significance level ($p < 1.39 \times 10^{-7}$, electronic supplementary material, figure S4). The significant SNPs, that were located on four different scaffolds of the Arctic fox genome (electronic supplementary material, table S14), were BLASTed against the annotated dog genome *CanFam 3.1*. We obtained a match in the dog genome for 489 SNPs (486 on chromosome 5 (figure 1*a*), two on chromosome 27 and one on chromosome 17, electronic supplementary material, table S14). The BLAST results also show that the four scaffolds which mapped to chromosome 5 and contain significant SNPs assemble next to each other (figure 1*a*). A total of 438 SNPs were intragenic in the dog genome, whereas the remaining 51 SNPs were located in intergenic regions. The intragenic SNPs were distributed across 97 different genes (electronic supplementary material, table S15). An additional 57 genes were found less than 20 kb away from a significant SNP, with 34 of these being closer than 10 kb from a significant SNP (electronic supplementary material, table S15). The positions of significant SNPs on chromosome 5 stretched from 52 617 594 to 76 592 936 bp (figure 1*a*), a distance that appears to be longer than that of strong LD in the Arctic fox genome (electronic supplementary material, figure S12). A total of 379 genes are located in this region of the dog genome.

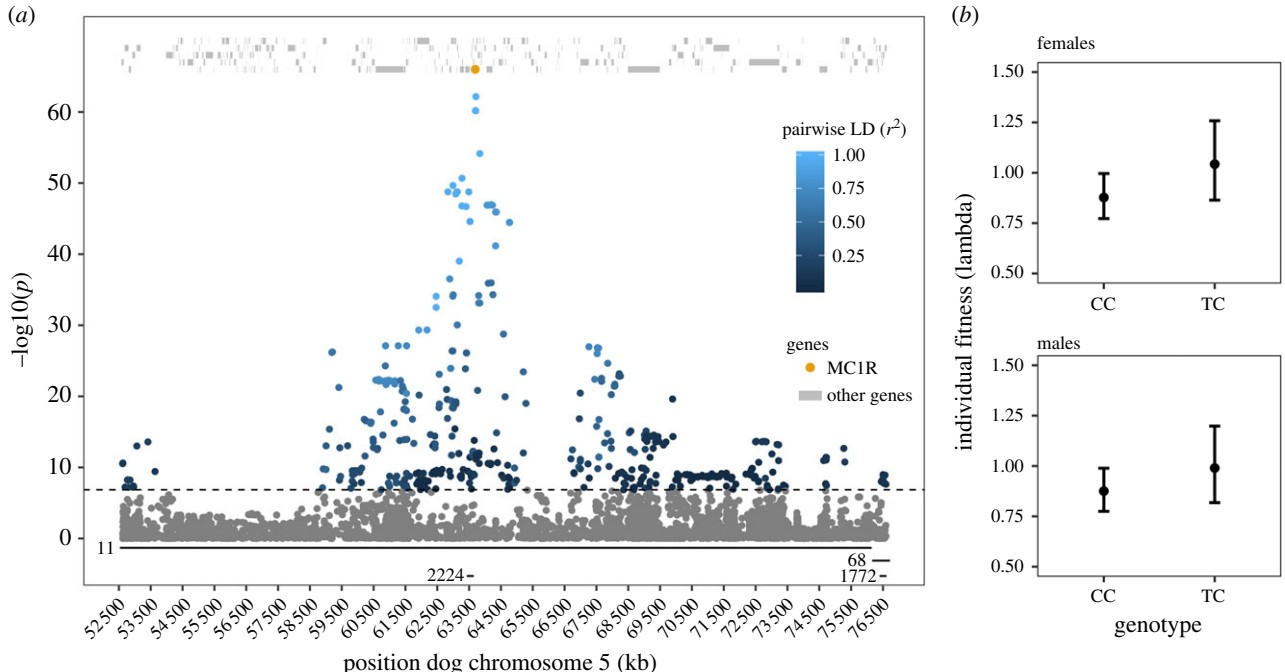

**Figure 1.** (*a*) Plot showing BLAST determined dog chromosome 5 locations of 486 SNPs significant in GWAS of fur colour in Arctic fox. The horizontal lines above the *x*-axis and the corresponding numbers show how the different Arctic fox scaffolds BLAST to dog chromosome 5. On the *y*-axis, significance levels of the SNPs in the GWAS are shown on a negative log scale. Pairwise LD ($r^2$) between top SNP AX177333963 and the other significant SNPs is shown by the blue colour gradient. All dog genes in the region are shown as grey lines at the top. The position of putative causal gene *MC1R* is shown with an orange dot (note that the *y*-axis values do not apply for genes). The dashed horizontal line shows the significance threshold after Bonferroni correction of the GWAS. (*b*) Predicted fitness (lambda) of the Arctic fox fur colour morph genotypes CC (white) and TC (blue). Whiskers represent 95% confidence intervals of predicted values. Predictions are based on additive GLMMs with genotype as predictor variable and year and subpopulation as random factors. (Online version in colour.)

The Affymetrix SNP array used in this study did not include any SNP located in the intragenic region of the candidate gene *MC1R*. SNP AX-176934441 was the closest significant SNP (5961 bp upstream; $p = 6.7 \times 10^{-61}$) and was chosen as the diagnostic SNP for the alternative genotypes at the *MC1R* gene in further analyses. Indeed, there was a near-perfect Mendelian relationship between genotypes at *MC1R* and fur colour phenotypes, where the C allele represented a recessive white fur colour allele, and T a dominant blue fur colour allele (878 of 882 CC individuals were white, 221 of 234 TC individuals were blue and nine of nine TT individuals were blue; electronic supplementary material, figure S6 and table S4). The *MC1R* genotypes agree with simple Mendelian inheritance of fur colour phenotype for 98.4% of the 681 Arctic foxes that were genotyped at the Affymetrix SNP array (electronic supplementary material, table S4). Furthermore, genotyping of 444 Arctic foxes with fur colour phenotype on *MC1R* using a Fluidigm SNP array confirmed this result: genotypes of 98.6% individuals were concordant with a simple Mendelian mode of inheritance (electronic supplementary material, table S4). Analysis of 12 whole-genome sequenced Arctic fox individuals (11 white and 1 blue) found the same base-pair mutations in *MC1R*, that were found previously [29], in the one blue individual. The 11 white individuals did not show these mutations. All other SNPs found in the *MC1R* sequence had the same genotype in one or more white foxes and the blue fox. See the electronic supplementary material, S12 for detailed information.

## (b) Selection analyses

Owing to low sample size, TT individuals had to be excluded from the analyses of individual fitness ($n_{\text{females}} = 2$, $n_{\text{males}} = 5$).

Annual individual fitness appeared higher for heterozygous (TC) females than females homozygous (CC) for the white allele, although not statistically significant at the 0.05 level ($b_{\text{TC}} = 0.173 \pm 0.102$, 95% confidence interval (CI) (−0.030, 0.370), $\chi^2_1 = 2.79$, $p = 0.095$, figure 1*b*; electronic supplementary material, table S8). The same pattern was present in males ($b_{\text{TC}} = 0.123 \pm 0.105$, 95% CI (−0.086, 0.325), $\chi^2_1 = 1.35$, $p = 0.245$, figure 1*b*; electronic supplementary material, table S8). The effects of genotype on individual fitness were found to be independent of origin, rodent phase and snow fall (electronic supplementary material, table S12).

In the analysis of fitness components, heterozygous individuals were found to have higher fecundity than homozygote CC individuals ($b_{\text{TC}} = 0.497 \pm 0.162$, $\chi^2_1 = 4.54$, $p = 0.033$). This effect tended to be more pronounced in females than in males (genotype x sex interaction: $\chi^2_2 = 5.47$, $p = 0.065$; figure 2*a*; electronic supplementary material, table S9). In addition, the difference between the two fur colour genotypes in fecundity was more pronounced in years of low (i) and increasing rodent phase (ii), where adult TC individuals produced more recruits than CC individuals ($\chi^2_3 = 9.32$, $p = 0.025$; electronic supplementary material, figure S9b). Differences in the effects of genotype on fecundity did not depend on origin and snowfall (electronic supplementary material, table S12).

Survival tended to be higher for heterozygous individuals compared to homozygous CC individuals ($b_{\text{TC}} = 0.296 \pm 0.157$, $\chi^2_1 = 3.632$, $p = 0.057$, figure 2*b*; electronic supplementary material, table S10), with no difference between sexes (genotype × sex interaction: $\chi^2_2 = 0.4321$, $p = 0.8057$). There was also a tendency for the difference in survival between genotypes to depend on rodent phase ($\chi^2_3 = 7.36$, $p = 0.061$; electronic supplementary material, figure S9a), where

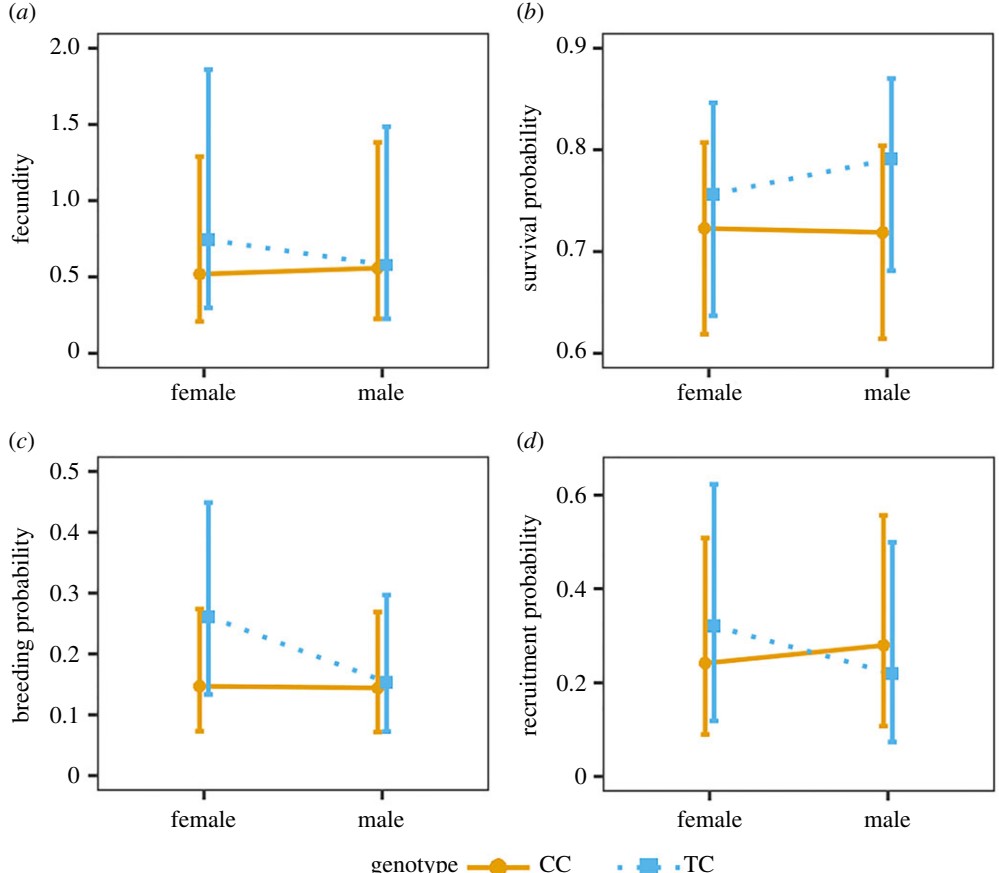

**Figure 2.** Predicted fecundity (number of recruits produced (a), adult survival probability (b), adult breeding probability (c) and juvenile recruitment probability (d)) for female and male Arctic foxes with fur colour genotypes CC (white) and TC (blue). Whiskers represent 95% confidence intervals of predicted values. Predictions are based on GLMMs with genotype, sex and their interaction (genotype × sex) as predictor variables and year and subpopulation as random factors. (Online version in colour.)

heterozygous individuals had higher survival than homozygous individuals in low (i) and increasing (ii) rodent phase. Differences in the effects of genotype on survival did not depend on origin and snowfall (electronic supplementary material, table S12).

An individual's probability of breeding was found to be significantly higher for heterozygous than homozygous females ($\chi_1^2 = 5.949$, $p = 0.015$), but there was no difference between the two genotypes in males (genotype × sex interaction: $\chi_2^2 = 7.487$, $p = 0.024$; figure 2c; electronic supplementary material, table S11).

All three adult fitness components (fecundity, survival probability and breeding probability) first increased with age and then decreased at older ages (electronic supplementary material, figure S8), but there were no differences in the effects of genotypes between age classes (i.e. no significant genotype × age interactions; electronic supplementary material, S7).

The recruitment probability (juvenile survival) was found to be independent of genotype ($\chi_1^2 = 0.012$, $p = 0.914$, figure 2d), did not depend on sex ($\chi_1^2 = 0.004$, $p = 0.948$), and the lack of any relationship between genotype and recruitment probability was similar in both sexes (genotype × sex interaction: $\chi_1^2 = 2.403$, $p = 0.121$). Hence, the higher fecundity of heterozygous individuals originated from the probability of breeding and/or the number of pups produced.

Genome-wide heterozygosity was $0.020 \pm 0.007$ lower in wild-born foxes ($n = 312$) compared to foxes born at the breeding station ($n = 374$), and foxes ($\chi_1^2 = 5.34$, $p = 0.021$, $n = 374$), and foxes

heterozygous at *MC1R* ($n = 123$) had $0.006 \pm 0.002$ higher genome-wide heterozygosity than foxes with the CC genotype ($\chi_1^2 = 5.96$, $p = 0.015$, $n = 563$). The higher genome-wide heterozygosity of TC foxes was similar for individuals of different origin (genotype × origin interaction: $\chi_1^2 = 0.14$, $p = 0.710$). Individual fitness and the fitness components investigated did however not depend on genome-wide heterozygosity (electronic supplementary material, table S13).

The candidate region GWAS for individual fitness revealed one SNP significantly associated with individual fitness at the Bonferroni-corrected significance level ($p < 1.24 \times 10^{-5}$, electronic supplementary material, figure S10). However, this SNP (AX-177107035, $p = 3.56 \times 10^{-6}$) was not significantly associated with Arctic fox fur colour.

## (c) Gene analyses

Many genes are located close to and/or are in strong LD with *MC1R* (figure 1a). Consequently, changes in *MC1R* allele- or genotype frequencies would lead to changes in frequencies of variants at other genes as well. To gather insight on what functions these genes have and how they might affect Arctic foxes, we conducted some preliminary GO investigations. For 132 genes that were found to be less than 10 kb away from an SNP significantly associated with Arctic fox fur colour, a GO term enrichment analysis showed over-representation of 33 GO terms (electronic supplementary material, table S16). Many of these GO terms represent fundamental biological functions (e.g. cytoplasm, intracellular or organelle). Eight of

the 33 over-represented GO terms are involved in metabolic processes, six of them in lipid metabolism (electronic supplementary material, table S16). Other GO terms are involved in developmental processes (developmental processes, regulation of Wnt signalling pathway).

To limit the analysis to genes that probably are highly associated with *MC1R* genotype, we looked for genes closer than 10 kb to an SNP that is (i) significantly associated with Arctic fox fur colour and (ii) in high LD ($r^2 \geq 0.5$) with the SNP most associated with Arctic fox fur colour. Here, 41 genes were found, and their functions according to UniProtKB are summarized in the electronic supplementary material, table S17. Only three of these genes were Swiss-Prot reviewed for dogs, *MC1R* being one of them. For several of the genes listed here, important functions are known. These include regulation of the Wnt signalling pathway (*CTNNBIP1*), DNA reparation (*FANCA*), glucose metabolism (*H6PD*), development (*RERE*) and immune response (*PIK3CD*, *BANP*). These 41 genes were included in the GeneMANIA analysis, which showed co-expression of *MC1R* with four genes: *CTNNBIP1*, *GSE1*, *PIEZO2*, *TCF25* (electronic supplementary material, figure S11). One gene (*HSBP1*) was located closer than 20 kb to the SNP that was significantly associated with individual Arctic fox fitness (electronic supplementary material, figure S10). *HSBP1* plays a role in stress resistance and actin organization.

## 4. Discussion

In this study, we investigated the genetic basis and architecture of Arctic fox fur colour. Our results demonstrate that *MC1R* is the only causal gene underlying the white and blue fur colour morphs in the Arctic fox. Quantification of selection on the colour morphs showed signs of a fitness advantage of heterozygous individuals at the fur colour locus that appeared to be similar across most environmental conditions. This fitness advantage was stronger in females than in males, and different fur colour genotypes were to some extent affected differently by food access (rodent cycle). The *MC1R* gene is located in a gene-rich region in the Arctic fox genome, and gene analyses showed that SNPs in several genes involved in developmental and metabolic processes are in strong LD with the diagnostic Arctic fox fur colour SNP.

Our GWAS identified many SNPs with significant association with Arctic fox fur colour (figure 1*a*; electronic supplementary material, figure S4). BLAST results showed that all but three of the significant SNPs were near *MC1R* in the region from 52 to 77 Mb on dog chromosome 5 (figure 1*a*). Our results also showed that the different scaffolds that contain significant SNPs assemble next to each other (figure 1*a*; electronic supplementary material, table S14) and thus do not represent independent peaks of significance. Together with *MC1R* sequence data from 12 whole-genome sequenced Arctic foxes (electronic supplementary material, S12) and the near-perfect association between Arctic fox fur colour and the top SNP genotype, these GWAS results support the hypothesis by Adalsteinsson *et al.* [30] that fur colour morphs in Arctic fox is determined by a single Mendelian gene and the results of Våge *et al.* [29] which suggested *MC1R* as the sole causal gene underlying the distinct Arctic fox fur colour morphs. The few cases of mismatch between recoded fur colour phenotype and expected genotype are likely to be caused by incorrect phenotyping in the field as field data are collected under sometimes demanding conditions. The genome-wide scale of this study confirms that no other areas in the genome explained variation in fur colour and provides firm evidence of *MC1R*'s role based on much larger sample size than previously applied. While the candidate gene approach has worked in this case, large-scale genome scans should be the preferred method to verify causal genes owing to their unbiased approach [38].

The quantification of selection showed that foxes heterozygous at the fur colour locus tend to have higher individual fitness than individuals that are homozygous for the white allele C (figure 1*b*). Our analyses revealed a larger difference in fitness in female Arctic foxes than in male individuals (figure 1*b*). Decomposition of fitness into different components corroborated the results based on individual fitness and showed that TC individuals (blue) scored better in fecundity, breeding probability and adult survival probability than CC individuals (white), with the effects on the fecundity measures being strongest in female foxes (figure 2). The fitness differences between the two fur colour genotypes were more pronounced at low and increasing rodent abundance (electronic supplementary material, figure S9). These results suggest that blue foxes have a higher probability than white foxes to survive under poor food conditions and show a stronger ability to use favourable conditions for reproduction (i.e. during years of increasing rodent abundance where juvenile survival and subsequent recruitment is high) [68].

Unfortunately, blue homozygotes (TT) were rare in the study area. Thus, the differences between blue homozygotes and the other genotypes could not be reliably quantified. The frequency of blue homozygotes is increasing in the Scandinavian Arctic fox population, hence such analyses may be possible in the future.

Evolutionary mechanisms underlying Arctic fox fur coloration are not well studied and the main difference between the colour morphs is thought to be their camouflage value in different habitats [34]. Recently, Di Bernardi *et al.* [69] also showed fitness advantages in Norwegian blue foxes. However, the performance of the two colour morphs was not differentially affected by the tested climatic variables (snow cover and winter temperature), except for a weak indication of thermal advantage of blue juveniles, with a tendency of higher survival in colder winter temperatures compared to white juveniles [69]. Overall, they did not find consistent evidence that these advantages are owing to differences in camouflage or thermoregulation [69]. Likewise, our results are not plausibly explained by differences in camouflage (i.e. white morph is expected to have better camouflage values in mountain habitats [34]) or thermoregulation (i.e. start and end of snow season did not affect fitness of colour genotypes differently (electronic supplementary material, table S12)), indicating that that the adaptations the Arctic fox has to withstand the Arctic winter are to a large extent independent of colour genotype.

Because support for the two most likely routes of direct selection on Arctic fox fur colour is weak or missing in our study, it seems reasonable to explore potential routes of indirect selection. Pleiotropic interactions in the melanocortin complex, which *MC1R* is part of, are well known and reviewed [26]. Both experimental and observational studies have shown a large variety of traits that are affected by the

melanocortin system and thus covary with melanin-based coloration [26]. These traits, e.g. resistance to stressors and enhanced immune response, have the potential to play vital roles for a wild species living in a harsh climate [26]. Behavioural traits such as aggressiveness are affected by pleiotropy in the melanocortin system and could impact foxes with genotypes for the blue colour morph positively in terms of getting access to good den sites and chasing away both conspecifics as well as competitors (e.g. red foxes). The last group of traits affected by pleiotropy in the melanocortin system is sexual traits, where both sexes can be affected positively in terms of sexual receptivity and fertility [26]. One could also expect higher fertility in male blue foxes based on findings that male blue foxes had higher concentrations of spermatozoa in their ejaculates compared to white foxes [70]. Yet, our findings do not concur as we did not find a difference in fecundity or breeding probability between male foxes with the CC (white) and TC (blue) genotype (figure 2a,c), indicating that any difference in spermatozoa concentration does not translate into higher reproduction in wild Arctic foxes in the Scandinavian population.

*MC1R* is located in a region with numerous other genes and we found several genes close to SNPs that were significantly associated with Arctic fox fur colour (figure 1a). Based on an analysis of LD (figure 1a; electronic supplementary material, figure S12), some of these genes certainly covary with fur colour genotype in the Scandinavian Arctic fox. Both the GO term analysis (electronic supplementary material, table S16) and the analysis of genes close to SNPs significantly associated with Arctic fox fur colour (electronic supplementary material, table S17) show that genes covarying with *MC1R* genotypes may be involved in important processes. As for all species enduring harsh winter conditions, the ability to control metabolism is relevant and potentially vital for Arctic foxes in enduring cold climate and food scarcity. Eight of the over-represented GO terms were related to lipid and steroid metabolism, making this an interesting pathway to investigate for future studies. Regulation of the Wnt signalling pathway showed up in our results as an enriched GO term (electronic supplementary material, table S16), as well as a single gene in form of *CTNNBIP1*. This pathway plays significant roles in organism development [71] and inhibition can lead to severe and potentially fatal effects [72]. Several other over-represented GO terms were also part of developmental processes (electronic supplementary material, table S16). In addition, the gene *RERE* that plays a role in developmental processes was found among the genes likely to covary with *MC1R*. Another two of these genes are involved in immune responses (*BANP* and *PIK3CD*), a trait that also is part of the pleiotropic melanocortin system. *HSBP1* was found close to the SNP significantly associated with individual Arctic fox fitness and is involved in stress resistance. Precisely how these genotypes are expressed phenotypically and whether these phenotypes can affect fecundity and/or viability in the Arctic fox remains to be seen. However, although there is a risk of 'storytelling' in this kind of analysis [73], these genes provide examples of covarying genes that may potentially have implications for individual Arctic fox fitness and should be investigated in more detail in future studies that aim to understand the molecular basis for fitness differences between Arctic fox fur colour genotypes and phenotypes.

Another possible explanation of higher fitness in blue heterozygotes compared to white homozygotes is

heterozygosity advantage [58,59]. However, despite having found significantly higher genome-wide heterozygosity in individuals heterozygous at the fur colour locus compared to those homozygous for the C allele, variation in individual fitness did not seem to be driven by this difference (electronic supplementary material, table S13). We also showed that foxes born at the captive breeding station had higher genome-wide heterozygosity than individuals born in the wild. This could indicate a lower degree of inbreeding in captive-born foxes, which may seem counterintuitive at first glance. However, this is expected as breeding pairs in the breeding station represent all extant subpopulations in Scandinavia and are chosen to maintain genetic diversity [40]. Hasselgren *et al.* [74] presented a good example of the genetic rescue effect where blue Arctic foxes showed high reproductive success in an inbred population in Sweden. It is possible that we see a weak genetic rescue effect in this study as well, and that the observed reproductive advantages of heterozygous individuals (figures 1b and 2) might be the result of genetic rescue by the release of TC individuals from the breeding station. The importance of such effects, and whether they contribute to explaining the observed growth of the Scandinavian Arctic fox population, will be explored in future studies.

Our study adds to the body of research that has identified major genes underlying traits with fitness implications for a wild animal species through genetic mapping [15–18]. However, our results also reveal the large potential for interesting genetic interactions that are hidden behind the seemingly simple trait, such as Arctic fox fur colour. Covariation between colour and other phenotypic traits is well documented [26,27] and our overall results may suggest that such covariation, owing to LD between *MC1R* and other genes or pleiotropic effects of *MC1R*, may be the driver of selection on fur colour also in the Scandinavian Arctic fox population. This emphasizes the need to look further than the most apparent phenotype when attempting to understand the mechanisms of selection in wild populations. More specifically, it is clear that gene mapping can provide valuable insight into the genetic architecture of adaptive traits and other linked traits. Also, when the linked gene that actually affects individual fitness cannot be identified, the molecular genetic information generated (e.g. on linked genes, pleiotropy, genome-wide heterozygosity) can be used to determine knowledge gaps and areas of interest for future research. In our study species, one major issue is the lack of data on other phenotypic differences between the colour morphs, such as behaviour, metabolism, energy expenditure or immune response. Future research may use such traits as a starting point for gaining more insight into selection processes that occur in the Arctic fox.

**Ethics.** Necessary permits for research on a wild species in Norway were in place and all research was conducted according to national rules and regulations. Permits include approvals of animal care protocols for captive breeding and live capture of wild animals, as well as permits for conducting research and handling a species of conservation concern.

**Data accessibility.** Because the Arctic fox is an endangered species in the study area, sensitive data (i.e. den locations) will not be released. Owing to ongoing research projects and the conservation status of the Scandinavian Arctic fox we ask for a 2 year embargo before making data available.

**Authors' contributions.** L.T.: conceptualization, data curation, formal analysis, investigation, methodology, project administration, software, validation, visualization, writing—original draft,

writing—review and editing; I.J.H.: conceptualization, data curation, formal analysis, methodology, supervision, writing—original draft, writing—review and editing; O.K.: data curation, supervision, writing—original draft; C.D.B.: data curation, investigation, methodology, writing—original draft, writing—review and editing; T.K.: formal analysis, methodology, supervision, writing—original draft, writing—review and editing; K.N.: data curation; M.H.: data curation; J.F.W.: data curation; A.A.: data curation; A.L.: conceptualization, data curation, funding acquisition, project administration, supervision, writing—original draft, writing—review and editing; N.E.E.: conceptualization, data curation, funding acquisition, investigation, methodology, project administration, writing—original draft, writing—review and editing; Ø.F.: conceptualization, data curation, funding acquisition, investigation, project administration, supervision, writing—original draft, writing—review and editing; H.J.: conceptualization, funding acquisition, investigation, project administration, supervision, writing—original draft, writing—review and editing. All authors gave final approval for publication and agreed to be held accountable for the work performed therein.

Competing interests. We declare we have no competing interests.

Funding. The Norwegian Captive Breeding Programme (contract 19087015) and the Arctic Fox Monitoring Programme in Norway (contract 18087019) contributed a significant amount of data to this study, all funded by the Norwegian Environmental Agency. This study was part of ECOFUNC funded by the Research Council of Norway (grant 244557). The work was also partly supported by the Research Council of Norway through its Centres of Excellence funding scheme (grant 22357).

Acknowledgements. We like to thank numerous park rangers and summer field assistants involved in collecting field data and genetic samples. Thanks to many people at the Genlab at NINA for performing genetic analysis and to Roger Meås and Roy Anderson for operating chip readers and all technical equipment. Further we would like to thank Kristine Ulvund who extracted and merged recovery and reproduction data, Stefan Blumentrath for extracting snow data, Alina Niskanen for help with genetic parentage analyses and Sarah Lundregan and Martin Kuiper for fruitful discussions about GWAS, BLAST and GO analyses. Genotyping on the custom fox Affymetrix Axiom 702 K SNP array was carried out at CIGENE, Norwegian University of Life Sciences, Norway.

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
