## [Peer Review File · Proceedings of the Royal Society B: Biological Sciences]

Review History

RSPB-2020-1850.R0 (Original submission)

Review form: Reviewer 1

Recommendation

Accept with minor revision (please list in comments)

Scientific importance: Is the manuscript an original and important contribution to its field?

Acceptable

General interest: Is the paper of sufficient general interest?

Good

Quality of the paper: Is the overall quality of the paper suitable?

Good

Is the length of the paper justified?

Yes

Should the paper be seen by a specialist statistical reviewer?

Yes

Do you have any concerns about statistical analyses in this paper? If so, please specify them explicitly in your report.

No

It is a condition of publication that authors make their supporting data, code and materials available - either as supplementary material or hosted in an external repository. Please rate, if applicable, the supporting data on the following criteria.

Is it accessible?

No

Is it clear?

N/A

Is it adequate?

No

Do you have any ethical concerns with this paper?

No

Comments to the Author

This paper presents genome-wide association of fur colour as a binary trait in Arctic foxes, and quantifies the fitness effects of this locus. The result confirms a previous candidate gene association at MC1R, and suggests that blue colour is associated with higher fitness.

The genomic methods for mapping are appropriate, and the paper is well-written. I have to declare that I don't have the expertise to evaluate the details of the demographic model of fitness.

I have a few comments about the use of Gene Ontology enrichment to analyse the potential for pleiotropy, the bioinformatics of mapping significant loci to chromosomes, and data availability.

Gene Ontology enrichment as potential for pleiotropy

The paper discusses the potential importance of pleiotropy and hypothesises about different causes for correlated effects (either other genes are in LD or dragged along by selection on MC1R (lines 399-422), or there are functional interactions with other melanocortin-regulated traits (lines 94-95)). However, the only evidence of pleiotropy is gene ontology enrichment, meaning that genes of certain annotations are close to MC1R in the dog genome. Thus the Gene Ontology analysis, in my opinion, needs to be treated much more cautiously.

Gene Ontology enrichments applied to regions of the genome are known to be subject to false positives (Pavlidis et al. 2012 <https://academic.oup.com/mbe/article/29/10/3237/1032149>). Furthermore, there is the added uncertainty of mapping genes between fox and dog, as well the uncertainty in the annotation (e.g., the first term listed in the supplementary table of enriched GO terms has been declared obsolete by the GOA).

Matches to more than one chromosome

The BLAST search and Manhattan plot (supplementary figure 3) show significant associations to more than one chromosome. It would be good to make clear the reasoning for discarding the BLAST hits to chromosomes 27 and 17, and the evidence that the hits are all from the same locus.

On that note, I find the Supplementary figure 3 quite convincing, and I think it could be part of Figure 1. I concede that this probably is a matter of personal taste.

Accuracy of fitness measurement

The supplementary analysis with known pedigree suggests that parentage assignment is accurate. However, I am left wondering how certain we can be that the pedigree (from the wild animals) is complete i.e. how accurate is the estimate of the number of pups? Is this obvious from the quality of the monitoring data, or is there room for uncertainty? As far as I can tell, this is not addressed in the paper.

Data and code availability

The reviewer instructions specify that we must check all supplementary material to ensure that all data has been submitted. It appears that the underlying data was not available for review.

The paper states that data and scripts will be made available on Dryad. It would have been useful to see these as a reviewer, especially since the verbal descriptions of the models (lines 193-200) are not easy to follow, even with the aid of the Supplementary materials.

Minor comments

Introduction, lines 68-72: This passage seems important for the framing of the study, but it is hard to follow exactly what it says and why it "demonstrates the importance of more studies ... into the genetic architecture of adaptive traits". Is the point that multivariate selection on correlated traits complicates prediction of the selection response? How does this study contribute to illuminating this question?

Lines 93-94: I agree that candidate gene approaches have serious pitfalls. However, it also seems to have worked reasonably well for pigmentation in multiple species, including this one; this study finds, indeed that the expected candidate gene is a major locus for the trait.

Methods, lines 117-119: Some more detail about the scat sampling would be good. Did those data include fur colour phenotypes?

Figure 1b and 2: Wouldn't confidence intervals for the difference in fitness be more useful than 95% prediction intervals?

Review form: Reviewer 2 (Anna Santure)

Recommendation

Major revision is needed (please make suggestions in comments)

Scientific importance: Is the manuscript an original and important contribution to its field?

Excellent

General interest: Is the paper of sufficient general interest?

Excellent

Quality of the paper: Is the overall quality of the paper suitable?

Good

Is the length of the paper justified?

Yes

Should the paper be seen by a specialist statistical reviewer?

No

Do you have any concerns about statistical analyses in this paper? If so, please specify them explicitly in your report.

Yes

It is a condition of publication that authors make their supporting data, code and materials available - either as supplementary material or hosted in an external repository. Please rate, if applicable, the supporting data on the following criteria.

Is it accessible?

No

Is it clear?

No

Is it adequate?

No

Do you have any ethical concerns with this paper?

No

Comments to the Author

Thank you for the opportunity to review this paper, it has been a pleasure to read and to work through the analyses. It has been particularly impressive to see such a comprehensive treatment of the fitness consequences of the diagnostic SNP for the presumed MC1R 'causative' haplotype, and I enjoyed the discussion of the possibility of effects of other nearby genes via linkage and pleiotropy. I also think it is great that you've used a robust statistical approach to test for association to be able to confirm previous analyses. The writing is clear and analyses are generally well explained (although my apologies, I was very confused by the fitness analyses!). For clarity I think it would have been nice to include some more of the supplementary information in the manuscript, but I appreciate that the page limits are rather restrictive which does unfortunately force brevity in the methods and results sections. I have a few general comments and questions and some smaller edits. Please note that despite the length of my comments I do think this is an excellent manuscript and I commend you on some fantastic science!

The first general comment is around the choice to merge datasets across the range to test for association. I appreciate that this was done to increase sample size to detect power for association, but I am a little worried at the high genomic inflation (you do not provide a Q-Q plot to show the distribution of p-values - please do so in your supplement -, but they clearly seem to be skewed by more than simply your significant peaks). From my understanding your GWAS model includes only the GRM as a random effect and there are no other fixed or random effects fitted? This is possible to do in RepeatABEL (even if you don't have repeat measures) and we found Lars Ronnegard incredibly helpful when we fitted our hihi GWAS (<https://doi.org/10.1098/rspb.2020.0948> - the dryad link for this paper has R code for fitting a model with multiple random and fixed effects in RepeatABEL) or I can also offer some pointers! I'd recommend using a structure-like analysis to test whether you have very clear structure in your dataset across subpopulations, and include subpopulation as a fixed effect if so, and then also include the GRM to account for fine-scale within population structure. There is a loss of power from 'double fitting' this population structure but the GWAS peak is so clear I'd think the most likely outcome is that your three additional peaks might disappear, which may not be a bad thing... my feeling is that these are either misassembled scaffolds that should be part of 11, or are picking up some level of structure in your data that hasn't been adequately accounted for. I was also not sure what effect the inclusion of the breeding population had on your overall structure. It seems this is a very mixed population and that animals are captive bred and then released to many of the subpopulations, so some populations are more 'pure' than others and some might be

quite mixed. Perhaps the 'origin' should also then be added as a fixed effect to your GWAS?

The second general comment is to do with understanding the data a bit better, I think it is a pity to not have Figure S1 (or some version of) in the main text, this would be particularly helpful if white/blue proportions were represented for each subpopulation on the map. See my comment below with regard to supplementary table S1, it would have been really great to have the categories of colour, location, origin, sex broken down more as I found it hard to assess the evidence that you'd used to decide that neither camouflage or thermoregulation were likely selection pressures for colour morph. I appreciate that this information is likely more thoroughly explored in the paper that is in review, but I think a little more information is needed to put the analyses / conclusions of this manuscript in context.

The third comment is around the statistical models fitted to test overall fitness and components thereof and how to understand what you term predicted values in Figure 1 and 2. I know you have summary tables and extensive text in the supplement but I am quite confused by what effects were included in which models and exactly how many models were fitted. For example in terms of effects on the six measures of fitness (male fitness, female fitness, fecundity, survival, etc), the following terms are mentioned: rodent and snow phases as shown in Tables S3 and S8, genotype and sex (Tables S4-S7), age, age², origin (Table S8), interactions thereof (Table S8), subpopulation and year (Figure S6). Mentioned in the main text but nowhere in the supplement are random intercepts for birth year and den. Were all of these terms fitted together into a single model for each of the six fitness phenotypes? Or were models for male and female fitness different from models for the other four fitness phenotypes? Or perhaps two models per phenotype as you test models with and without interactions (Table S8). Or did each fitness phenotype have a series of models that independently tested environmental models from the models that included other effects such as sex? I think all of this would immediately be cleared up by a summary table that lists every single model fitted with the fixed and random effects, has sample sizes for the number of individuals and number of phenotypes that makes clear what are repeated measures, and perhaps has a column that shows where the results of this model can be found in terms of supplementary tables and figures.

A fourth point where I am confused is what is meant by lambda. I would think of this as a whole lifetime measure of reproductive success i.e. a single value per individual. You mention predicted lambda and I therefore interpreted this as a single standardised fitness measure for each individual, standardised for all the significant effects of origin, subpopulation etc. However, in the description of the fitness GWAS in the supplement you use RepeatABEL to be able to use repeat measures of lambda. What are repeat measures of lambda? Why the need for RepeatABEL? And why can't you use the predicted lambda from your selection analyses to run a GWAS on? Further, why isn't origin included in the fitness GWAS? Again, some of this might be cleared up by a simple table that, similar to your fitness table, lists clearly the GWAS models, what the phenotypes are, and where to find the results. I did wonder if you used a single predicted lambda for your phenotype in the GWAS rather than what seems to be annual breeding data (???) whether your pvalue landscape might have looked a bit different? I would think it would have been quite nice to do this genome-wide. This of course makes the assumption that fitness has a significant heritable component, which is appearing rather unlikely if the diagnostic colour SNP is nowhere near significant in this GWAS, despite the relationship between genotype and fitness. The hint at heterozygote advantage at the diagnostic SNP may suggest that the fitness GWAS should be fitted as a dominant model, but these are drastically underpowered, and I don't think would perform very well given your lack of TT individuals.

Fifth, is this really a single gene trait? What has happened with the 'incorrect' phenotypes where genotype does not match morph? Incorrect phenotyping or sample mixups perhaps? I would be surprised if it is just imperfect LD with the causative SNPs, but perhaps that is an option? Are these morphs intermediate in any way that might suggest a modifier locus? Are they only one sex that might even suggest a sex-specific modifier? Assuming they are not some artefact of the GWAS not accounting for population structure adequately, are they true different genomic

location from the main peak? Is there any sign that the additional scaffolds could be placed on dog chromosome 5? What about blast matches with non-significant nearby SNPs, even if the top SNPs are not blasting to chromosome 5, that might suggest these smaller scaffolds should be added to / inserted into scaffold 11? I think you need to discuss the phenotype mismatch options in light of the three other significant peaks in your GWAS.

Minor comments / edits

Main text

-line 58: I think you could be a bit more assertive here! "Colouration is one of the most conspicuous..."

-line 68-72: this seems to be written in a way that suggests that there isn't theory yet to make these predictions, but of course there is. I think the point you are (correctly) trying to make is that measuring traits, their relationship to fitness, and their correlations to each other, is challenging in wild populations. Perhaps rewrite a little to make this a bit clearer?

-line 74: could you say and describe exactly how many morphs, rather than 'multiple'? It made me wonder if some are perhaps not quite so distinct from each other? That could represent a simple genetic basis for white vs blue but some nuance in gradients between these morphs determined by other loci (see question above about the phenotype 'misclassifications' and additional scaffolds above)

-line 79: delete 'the' to read 'is the result of dominant allele effects' or perhaps 'is a result of the effect of a dominant allele'

-line 88: it felt like a descriptor was missing here e.g. extracellular / intracellular to read e.g. 'within the intracellular region of the...'

-line 94: is 'gene complex' the correct phrase here? I think of a gene complex as being interacting genes in a protein complex e.g. to make a heterodimer. I feel like 'gene family' might be what you meant, as I am pretty sure MC1R does not form a complex directly with any of these other MCxR proteins?

-line 112: is it correct, as indicated by line 115 "the breeding station" that there is only one captive facility? It would be good if this could be stated clearly. Also, I think it is important to briefly state that "breeding pairs in the breeding station represent all extant subpopulations in Scandinavia and are chosen to maintain genetic diversity" which is currently on line 430-431 but I think would be helpful here in the methods section instead / as well as in the discussion.

-line 117-119: I was not really sure why these foxes are mentioned? Since they have no phenotype and no fitness? Or do you mean they were included in some of the fitness calculations, just not the juvenile survival, because you could put them in the pedigree and hence calculate a fecundity measure for them for a given year?

-line 123: can you please confirm that only arctic fox specific SNPs were analysed? There are none that are polymorphic between arctic fox and red fox? Or if there are, these were excluded from these analyses?

-line 165: I didn't get a feeling from this about recapture rates. For example, how often do you miss finding a fox in a particular census year but locate it again the following year?

-line 169: I also didn't get a feeling for what proportion of individuals in the wild would have their breeding den located and pups genotyped? Line 173-174 you assume these individuals to have not bred, but how realistic is that? What impact might missing data have on the wild population?

-line 202: add 'a' to read 'is to a large extent'

-line 209: I was a bit confused by this section as I think of heterozygote advantage as being at a single locus. I think this would be less confusing if this was instead termed as heterozygosity advantage / inbreeding depression as 'heterozygosity' is the term used when talking about heterozygosity-fitness correlations.

-line 234: I was confused when I came to Figure S8 in the supplement as I wasn't sure where it came from. Can you add here that presumably most of this co-expression is from human / model organism data? Is it mammals only (or could be restricted to mammals only)?

-line 243-244: I found it very interesting that there were only three SNPs that aligned elsewhere than dog chromosome 5 in the genome. I would like to know which dog chromosome the scaffolds 68, 1772 and 2224 aligned to (this could be easily done by blasting all the SNP flanking

sequences from these scaffolds on to the dog genome). My guess is that some of these scaffolds might nest inside scaffold 11. If they don't, I really query whether you can say that MC1R is the 'sole causal gene' (see comments above). The peak on scaffold 68 looks particularly well supported by 13 SNPs and it would be nice to also zoom in to the genes through this region. The three and two SNPs on the other two scaffolds could possibly be misplaced, have you checked their second highest blast hit even if their first hit is not to dog chromosome 5? Or the blast hits of other nearby SNPs in the same scaffold that are not significant?

-line 346: I am not sure at the choice of the word 'covary'. I think you've taken this a shortcut to say that SNPs in these genes are in strong LD with the diagnostic SNP. This is not quite the same thing as the causative colour SNP being in LD with causative SNPs in other genes, which is what I think you imply by the term covary. I'd recommend a softening.

-line 355-356: see above, I am a bit unsure of the assertion that no other genes explain variation when (if) you have significant GWAS peaks on 3 other scaffolds

-line 365: delete 'both' as you list three phenotypes

-line 373: although results are explained, it would have been good to see this uploaded as supporting information to assess these conclusions

-line 377: I am not sure I see duration of snow fall tested in Table S8, you have first and last snowfall but not the difference between them. Have you tested for the effects of length of winter in your data?

Supplementary material

-Table S1: I appreciate the summary table is going to get very complicated to do this, but I would like to see a: split for sex, phenotype and origin (wild/captive) for each of the numbers in this table. I would also like to see the table ordered in the same order (i.e. by latitude) as in the map. This would allow readers to see (i) any systematic pattern in colour for location, (ii) any bias in male / female ratio for location, (iii) how the wild/captive breaks down for each morph. In addition, it would be good to have each sample listed with all of their accompanying information. I recognise that burrow sites are sensitive but surely they could all be grouped by subpopulation?

-Figure S3 and Supplementary Table 8: I really would like to see the mappings of your significant SNPs to positions on the dog genome. Please can you add columns with dog chromosome and position and arctic fox position to supplementary table 8. I really wondered where the scaffolds 68, 1772 and 2224 matched to in the dog genome, and further whether the 'gap' you see in Figure S3 is just a region of non-significant SNPs or whether this is a region of your assembly where there might be evidence of a clear genome rearrangement event between dog and fox (which incidentally might also be caused by a missassembly of your scaffold).

-section on parentage analysis, second-to-last sentence: could you please explain what you mean by correct dummy parents being assigned even if parents are ungenotyped? Do you mean perhaps that sibs cluster together?

-supplementary material 9 - fitness GWAS: you say that the GRM was based on all SNPs that passed QC, in the main text you state these were LD pruned, which is correct?

-Figure S7 vs statement in main text lines 308-311 "Bonferroni-corrected significance level ($p = 1.24E-05$)". The p-value threshold does not match between the graph and the main text and I cannot see any sign of a SNP 'close to significant' (line 310) on this plot. Please check.

-Supplementary material 11 - LD decay: I would have liked the dataset to have been pruned to remove close relatives before calculating LD (although I see the argument to keep them in as the GWAS is leveraging both family linkage and population LD to detect association). There are likely also population specific LD decays. I think you should just acknowledge clearly that all individuals were used and that this will include close family members and is across populations so readers have this clear in their mind.

Very best wishes,
Anna Santure

Decision letter (RSPB-2020-1850.R0)

23-Nov-2020

Dear Mr Tietgen:

I am writing to inform you that your manuscript RSPB-2020-1850 entitled "Fur colour in the arctic fox – genetic architecture and consequences for fitness" has, in its current form, been rejected for publication in Proceedings B.

This action has been taken on the advice of referees, who have recommended that substantial revisions are necessary. With this in mind we would be happy to consider a resubmission, provided the comments of the referees are fully addressed. However please note that this is not a provisional acceptance.

Sincerely,
Professor Hans Heesterbeek
<mailto:proceedingsb@royalsociety.org>

Associate Editor
Board Member: 1
Comments to Author:
Dear Mr. Tietgen,

Thank you for your submission to our special issue on Wild Quantitative Genetics in Proc R Soc B. Your manuscript has now been reviewed by two experts in the field. Both reviewers were positive about the manuscript and many of their concerns could be alleviated by rewriting. I do

agree with reviewer 2's suggestions that reanalyzing the data with different controls for population subdivision could improve the paper.

Reviewer(s)' Comments to Author:

Referee: 1

Comments to the Author(s)

This paper presents genome-wide association of fur colour as a binary trait in Arctic foxes, and quantifies the fitness effects of this locus. The result confirms a previous candidate gene association at MC1R, and suggests that blue colour is associated with higher fitness.

The genomic methods for mapping are appropriate, and the paper is well-written. I have to declare that I don't have the expertise to evaluate the details of the demographic model of fitness.

I have a few comments about the use of Gene Ontology enrichment to analyse the potential for pleiotropy, the bioinformatics of mapping significant loci to chromosomes, and data availability.

Gene Ontology enrichment as potential for pleiotropy

The paper discusses the potential importance of pleiotropy and hypothesises about different causes for correlated effects (either other genes are in LD or dragged along by selection on MC1R (lines 399-422), or there are functional interactions with other melanocortin-regulated traits (lines 94-95)). However, the only evidence of pleiotropy is gene ontology enrichment, meaning that genes of certain annotations are close to MC1R in the dog genome. Thus the Gene Ontology analysis, in my opinion, needs to be treated much more cautiously.

Gene Ontology enrichments applied to regions of the genome are known to be subject to false positives (Pavlidis et al. 2012 <https://academic.oup.com/mbe/article/29/10/3237/1032149>). Furthermore, there is the added uncertainty of mapping genes between fox and dog, as well the uncertainty in the annotation (e.g., the first term listed in the supplementary table of enriched GO terms has been declared obsolete by the GOA).

Matches to more than one chromosome

The BLAST search and Manhattan plot (supplementary figure 3) show significant associations to more than one chromosome. It would be good to make clear the reasoning for discarding the BLAST hits to chromosomes 27 and 17, and the evidence that the hits are all from the same locus.

On that note, I find the Supplementary figure 3 quite convincing, and I think it could be part of Figure 1. I concede that this probably is a matter of personal taste.

Accuracy of fitness measurement

The supplementary analysis with known pedigree suggests that parentage assignment is accurate. However, I am left wondering how certain we can be that the pedigree (from the wild animals) is complete i.e. how accurate is the estimate of the number of pups? Is this obvious from the quality of the monitoring data, or is there room for uncertainty? As far as I can tell, this is not addressed in the paper.

Data and code availability

The reviewer instructions specify that we must check all supplementary material to ensure that all data has been submitted. It appears that the underlying data was not available for review.

The paper states that data and scripts will be made available on Dryad. It would have been useful to see these as a reviewer, especially since the verbal descriptions of the models (lines 193-200) are not easy to follow, even with the aid of the Supplementary materials.

Minor comments

Introduction, lines 68-72: This passage seems important for the framing of the study, but it is hard to follow exactly what it says and why it "demonstrates the importance of more studies ... into the genetic architecture of adaptive traits". Is the point that multivariate selection on correlated traits complicates prediction of the selection response? How does this study contribute to illuminating this question?

Lines 93-94: I agree that candidate gene approaches have serious pitfalls. However, it also seems to have worked reasonably well for pigmentation in multiple species, including this one; this study finds, indeed that the expected candidate gene is a major locus for the trait.

Methods, lines 117-119: Some more detail about the scat sampling would be good. Did those data include fur colour phenotypes?

Figure 1b and 2: Wouldn't confidence intervals for the difference in fitness be more useful than 95% prediction intervals?

Referee: 2

Comments to the Author(s)

Thank you for the opportunity to review this paper, it has been a pleasure to read and to work through the analyses. It has been particularly impressive to see such a comprehensive treatment of the fitness consequences of the diagnostic SNP for the presumed MC1R 'causative' haplotype, and I enjoyed the discussion of the possibility of effects of other nearby genes via linkage and pleiotropy. I also think it is great that you've used a robust statistical approach to test for association to be able to confirm previous analyses. The writing is clear and analyses are generally well explained (although my apologies, I was very confused by the fitness analyses!). For clarity I think it would have been nice to include some more of the supplementary information in the manuscript, but I appreciate that the page limits are rather restrictive which does unfortunately force brevity in the methods and results sections. I have a few general comments and questions and some smaller edits. Please note that despite the length of my comments I do think this is an excellent manuscript and I commend you on some fantastic science!

The first general comment is around the choice to merge datasets across the range to test for association. I appreciate that this was done to increase sample size to detect power for association, but I am a little worried at the high genomic inflation (you do not provide a Q-Q plot to show the distribution of p-values – please do so in your supplement -, but they clearly seem to be skewed by more than simply your significant peaks). From my understanding your GWAS model includes only the GRM as a random effect and there are no other fixed or random effects fitted? This is possible to do in RepeatABEL (even if you don't have repeat measures) and we found Lars Ronnegard incredibly helpful when we fitted our hihi GWAS (<https://doi.org/10.1098/rspb.2020.0948> - the dryad link for this paper has R code for fitting a model with multiple random and fixed effects in RepeatABEL) or I can also offer some pointers! I'd recommend using a structure-like analysis to test whether you have very clear structure in your dataset across subpopulations, and include subpopulation as a fixed effect if so, and then also include the GRM to account for fine-scale within population structure. There is a loss of power from 'double fitting' this population structure but the GWAS peak is so clear I'd think the most likely outcome is that your three additional peaks might disappear, which may not be a bad thing... my feeling is that these are either misassembled scaffolds that should be part of 11, or are picking up some level of structure in your data that hasn't been adequately accounted for. I was also not sure what effect the inclusion of the breeding population had on your overall structure. It seems this is a very mixed population and that animals are captive bred and then released to

many of the subpopulations, so some populations are more 'pure' than others and some might be quite mixed. Perhaps the 'origin' should also then be added as a fixed effect to your GWAS?

The second general comment is to do with understanding the data a bit better, I think it is a pity to not have Figure S1 (or some version of) in the main text, this would be particularly helpful if white/blue proportions were represented for each subpopulation on the map. See my comment below with regard to supplementary table S1, it would have been really great to have the categories of colour, location, origin, sex broken down more as I found it hard to assess the evidence that you'd used to decide that neither camouflage or thermoregulation were likely selection pressures for colour morph. I appreciate that this information is likely more thoroughly explored in the paper that is in review, but I think a little more information is needed to put the analyses / conclusions of this manuscript in context.

The third comment is around the statistical models fitted to test overall fitness and components thereof and how to understand what you term predicted values in Figure 1 and 2. I know you have summary tables and extensive text in the supplement but I am quite confused by what effects were included in which models and exactly how many models were fitted. For example in terms of effects on the six measures of fitness (male fitness, female fitness, fecundity, survival, etc), the following terms are mentioned: rodent and snow phases as shown in Tables S3 and S8, genotype and sex (Tables S4-S7), age, age², origin (Table S8), interactions thereof (Table S8), subpopulation and year (Figure S6). Mentioned in the main text but nowhere in the supplement are random intercepts for birth year and den. Were all of these terms fitted together into a single model for each of the six fitness phenotypes? Or were models for male and female fitness different from models for the other four fitness phenotypes? Or perhaps two models per phenotype as you test models with and without interactions (Table S8). Or did each fitness phenotype have a series of models that independently tested environmental models from the models that included other effects such as sex? I think all of this would immediately be cleared up by a summary table that lists every single model fitted with the fixed and random effects, has sample sizes for the number of individuals and number of phenotypes that makes clear what are repeated measures, and perhaps has a column that shows where the results of this model can be found in terms of supplementary tables and figures.

A fourth point where I am confused is what is meant by lambda. I would think of this as a whole lifetime measure of reproductive success i.e. a single value per individual. You mention predicted lambda and I therefore interpreted this as a single standardised fitness measure for each individual, standardised for all the significant effects of origin, subpopulation etc. However, in the description of the fitness GWAS in the supplement you use RepeatABEL to be able to use repeat measures of lambda. What are repeat measures of lambda? Why the need for RepeatABEL? And why can't you use the predicted lambda from your selection analyses to run a GWAS on? Further, why isn't origin included in the fitness GWAS? Again, some of this might be cleared up by a simple table that, similar to your fitness table, lists clearly the GWAS models, what the phenotypes are, and where to find the results. I did wonder if you used a single predicted lambda for your phenotype in the GWAS rather than what seems to be annual breeding data (???) whether your pvalue landscape might have looked a bit different? I would think it would have been quite nice to do this genome-wide. This of course makes the assumption that fitness has a significant heritable component, which is appearing rather unlikely if the diagnostic colour SNP is nowhere near significant in this GWAS, despite the relationship between genotype and fitness. The hint at heterozygote advantage at the diagnostic SNP may suggest that the fitness GWAS should be fitted as a dominant model, but these are drastically underpowered, and I don't think would perform very well given your lack of TT individuals.

Fifth, is this really a single gene trait? What has happened with the 'incorrect' phenotypes where genotype does not match morph? Incorrect phenotyping or sample mixups perhaps? I would be surprised if it is just imperfect LD with the causative SNPs, but perhaps that is an option? Are these morphs intermediate in any way that might suggest a modifier locus? Are they only one sex that might even suggest a sex-specific modifier? Assuming they are not some artefact of the

GWAS not accounting for population structure adequately, are they true different genomic location from the main peak? Is there any sign that the additional scaffolds could be placed on dog chromosome 5? What about blast matches with non-significant nearby SNPs, even if the top SNPs are not blasting to chromosome 5, that might suggest these smaller scaffolds should be added to / inserted into scaffold 11? I think you need to discuss the phenotype mismatch options in light of the three other significant peaks in your GWAS.

Minor comments / edits

Main text

-line 58: I think you could be a bit more assertive here! "Colouration is one of the most conspicuous..."

-line 68-72: this seems to be written in a way that suggests that there isn't theory yet to make these predictions, but of course there is. I think the point you are (correctly) trying to make is that measuring traits, their relationship to fitness, and their correlations to each other, is challenging in wild populations. Perhaps rewrite a little to make this a bit clearer?

-line 74: could you say and describe exactly how many morphs, rather than 'multiple'? It made me wonder if some are perhaps not quite so distinct from each other? That could represent a simple genetic basis for white vs blue but some nuance in gradients between these morphs determined by other loci (see question above about the phenotype 'misclassifications' and additional scaffolds above)

-line 79: delete 'the' to read 'is the result of dominant allele effects' or perhaps 'is a result of the effect of a dominant allele'

-line 88: it felt like a descriptor was missing here e.g. extracellular / intracellular to read e.g. 'within the intracellular region of the...'

-line 94: is 'gene complex' the correct phrase here? I think of a gene complex as being interacting genes in a protein complex e.g. to make a heterodimer. I feel like 'gene family' might be what you meant, as I am pretty sure MC1R does not form a complex directly with any of these other MCxR proteins?

-line 112: is it correct, as indicated by line 115 "the breeding station" that there is only one captive facility? It would be good if this could be stated clearly. Also, I think it is important to briefly state that "breeding pairs in the breeding station represent all extant subpopulations in Scandinavia and are chosen to maintain genetic diversity" which is currently on line 430-431 but I think would be helpful here in the methods section instead / as well as in the discussion.

-line 117-119: I was not really sure why these foxes are mentioned? Since they have no phenotype and no fitness? Or do you mean they were included in some of the fitness calculations, just not the juvenile survival, because you could put them in the pedigree and hence calculate a fecundity measure for them for a given year?

-line 123: can you please confirm that only arctic fox specific SNPs were analysed? There are none that are polymorphic between arctic fox and red fox? Or if there are, these were excluded from these analyses?

-line 165: I didn't get a feeling from this about recapture rates. For example, how often do you miss finding a fox in a particular census year but locate it again the following year?

-line 169: I also didn't get a feeling for what proportion of individuals in the wild would have their breeding den located and pups genotyped? Line 173-174 you assume these individuals to have not bred, but how realistic is that? What impact might missing data have on the wild population?

-line 202: add 'a' to read 'is to a large extent'

-line 209: I was a bit confused by this section as I think of heterozygote advantage as being at a single locus. I think this would be less confusing if this was instead termed as heterozygosity advantage / inbreeding depression as 'heterozygosity' is the term used when talking about heterozygosity-fitness correlations.

-line 234: I was confused when I came to Figure S8 in the supplement as I wasn't sure where it came from. Can you add here that presumably most of this co-expression is from human / model organism data? Is it mammals only (or could be restricted to mammals only)?

-line 243-244: I found it very interesting that there were only three SNPs that aligned elsewhere than dog chromosome 5 in the genome. I would like to know which dog chromosome the

scaffolds 68, 1772 and 2224 aligned to (this could be easily done by blasting all the SNP flanking sequences from these scaffolds on to the dog genome). My guess is that some of these scaffolds might nest inside scaffold 11. If they don't, I really query whether you can say that MC1R is the 'sole causal gene' (see comments above). The peak on scaffold 68 looks particularly well supported by 13 SNPs and it would be nice to also zoom in to the genes through this region. The three and two SNPs on the other two scaffolds could possibly be misplaced, have you checked their second highest blast hit even if their first hit is not to dog chromosome 5? Or the blast hits of other nearby SNPs in the same scaffold that are not significant?

-line 346: I am not sure at the choice of the word 'covary'. I think you've taken this a shortcut to say that SNPs in these genes are in strong LD with the diagnostic SNP. This is not quite the same thing as the causative colour SNP being in LD with causative SNPs in other genes, which is what I think you imply by the term covary. I'd recommend a softening.

-line 355-356: see above, I am a bit unsure of the assertion that no other genes explain variation when (if) you have significant GWAS peaks on 3 other scaffolds

-line 365: delete 'both' as you list three phenotypes

-line 373: although results are explained, it would have been good to see this uploaded as supporting information to assess these conclusions

-line 377: I am not sure I see duration of snow fall tested in Table S8, you have first and last snowfall but not the difference between them. Have you tested for the effects of length of winter in your data?

Supplementary material

-Table S1: I appreciate the summary table is going to get very complicated to do this, but I would like to see a: split for sex, phenotype and origin (wild/captive) for each of the numbers in this table. I would also like to see the table ordered in the same order (i.e. by latitude) as in the map. This would allow readers to see (i) any systematic pattern in colour for location, (ii) any bias in male / female ratio for location, (iii) how the wild/captive breaks down for each morph. In addition, it would be good to have each sample listed with all of their accompanying information. I recognise that burrow sites are sensitive but surely they could all be grouped by subpopulation?

-Figure S3 and Supplementary Table 8: I really would like to see the mappings of your significant SNPs to positions on the dog genome. Please can you add columns with dog chromosome and position and arctic fox position to supplementary table 8. I really wondered where the scaffolds 68, 1772 and 2224 matched to in the dog genome, and further whether the 'gap' you see in Figure S3 is just a region of non-significant SNPs or whether this is a region of your assembly where there might be evidence of a clear genome rearrangement event between dog and fox (which incidentally might also be caused by a missassembly of your scaffold).

-section on parentage analysis, second-to-last sentence: could you please explain what you mean by correct dummy parents being assigned even if parents are ungenotyped? Do you mean perhaps that sibs cluster together?

-supplementary material 9 – fitness GWAS: you say that the GRM was based on all SNPs that passed QC, in the main text you state these were LD pruned, which is correct?

-Figure S7 vs statement in main text lines 308-311 "Bonferroni-corrected significance level ($p = 1.24E-05$)". The p-value threshold does not match between the graph and the main text and I cannot see any sign of a SNP 'close to significant' (line 310) on this plot. Please check.

-Supplementary material 11 – LD decay: I would have liked the dataset to have been pruned to remove close relatives before calculating LD (although I see the argument to keep them in as the GWAS is leveraging both family linkage and population LD to detect association). There are likely also population specific LD decays. I think you should just acknowledge clearly that all individuals were used and that this will include close family members and is across populations so readers have this clear in their mind.

Very best wishes,
Anna Santure

Author's Response to Decision Letter for (RSPB-2020-1850.R0)

See Appendix A.

RSPB-2021-1452.R0

Review form: Reviewer 2 (Anna Santure)

Recommendation

Accept with minor revision (please list in comments)

Scientific importance: Is the manuscript an original and important contribution to its field?

Excellent

General interest: Is the paper of sufficient general interest?

Excellent

Quality of the paper: Is the overall quality of the paper suitable?

Good

Is the length of the paper justified?

Yes

Should the paper be seen by a specialist statistical reviewer?

No

Do you have any concerns about statistical analyses in this paper? If so, please specify them explicitly in your report.

No

It is a condition of publication that authors make their supporting data, code and materials available - either as supplementary material or hosted in an external repository. Please rate, if applicable, the supporting data on the following criteria.

Is it accessible?

No

Is it clear?

N/A

Is it adequate?

N/A

Do you have any ethical concerns with this paper?

No

Comments to the Author

Thanks for the opportunity to review the revised version of this manuscript. The authors have done an excellent job of addressing mine and the other reviewer's comments and I appreciate the significant work that has gone into doing so. Below I list (i) a few quick comments in response to their cover letter / response to reviewers (reviewer comments that I haven't talked about below I think have been VERY well explained and resolved by the authors, so all I can say is thanks for all that work to do so!) and (ii) some very minor suggestions I came across on a final read through the manuscript:

(i) response to cover letter

-in their cover letter the authors acknowledge that a small error in calculating annual individual fitness meant that the p-value for female annual individual fitness was no longer significant. I appreciate the authors fixing this analysis (and changing text in the manuscript to acknowledge this) and agree with them that the change has not impacted the overall conclusions of the manuscript.

-Reviewer 1 comments

R1a: It would be good to make clear the reasoning for discarding the BLAST hits to chromosomes 27 and 17, and the evidence that the hits are all from the same locus.

-Although the authors explain their reasoning for discarding these hits in their response, I think this does need a quick mention somewhere in the manuscript files. Probably the best place for this is just a footnote to Supplementary Table S14 explaining that the SNPs also aligned to Chr 5 with slightly lower blast scores and so most likely do reside on Chr 5, but were conservatively discarded from further analysis (or something to that effect)

R1b: On that note, I find the Supplementary figure 3 quite convincing, and I think it could be part of Figure 1. I concede that this probably is a matter of personal taste.

-I liked the idea but not the implementation of the changes to Figure 1 in response to this comment. Scaffold 11 looks like it is rather two genes as per the top panels. Scaffold 2224 looks like a value of negative 2224 i.e. -2224. It isn't super clear which is Scaffold 68 vs 1772. How about putting 1772 on the same plane as 11 and 2224 on the same plane as 68, and putting the associated scaffold numbers consistently right and align-centred with the scaffold i.e.

```

11-----2224-----68-----1772-

```

Reviewer 2 (i.e. my!) comments

R2a: long-winded question about population structure

-great response, thanks! And thanks for trying the RepeatABEL model, interesting that it found more significant SNPs! Please just add the lambda values also to the plots you now give in the supplement, the lambda values for the first two are in the main text, but not the final one.

-I'd love to also see the MDS plot coloured by origin. I agree that your GenABEL model has accounted for whatever population structure might be there, but it would be nice just to have an idea of the overall population structure across the range. Not essential, though!

R2b: other long-winded questions about understanding the data a bit better, the fitness models, me misunderstanding lambda and the GWAS

-nothing to add, just a thanks for your detailed responses and the addition of the clear tables and text, and congrats on the publication!

R2c: my comment re original line 88 was more to ask about whether the MC1R variants were in the extracellular vs the intracellular vs the transmembrane domain of this trans-membrane protein. This is interesting because it has implications for what protein-MC1R interactions might be most likely to be messed up. But, no problem if you're not sure, it is a pretty minor detail!

R2d: I asked about the p-value threshold in Figure S7 (now Figure S4) and you replied that the threshold should be at $-\log(1.24e-5)=4.9$.

-Apologies, I put the wrong number in my question! What I was querying here was the difference in the p-value threshold between the colour GWAS and the fitness GWAS but I realised from your response that the Bonferonni thresholds are of course different because there are different numbers of SNPs tested (genome vs region). Perhaps this could be clarified in the caption for what is now Figure S10?

(ii) minor suggestions

-line 299: suggest "The BLAST results also show that the four scaffolds that mapped to chromosome 5 and contain significant SNPs assemble next to each other (Figure 1a).

-line 413: add "do" to read "and thus do not represent"

-Supplement page 8: can you please clarify how many SNPs were kept / discarded when you say "The analysis was restrained to SNPs lying on arctic fox scaffold 11 and that matched with a position on dog chromosome 5 during the BLAST". I am also not sure about the statement "SNPs on other scaffolds or that matched with different dog chromosomes would naturally appear as outliers in Figure S2" - first, do you rather mean Figure S4 (the GWAS) and second I am not sure 'outliers' is the right word here as SNPs within scaffold 11 that matched with another dog chromosome would not be highlighted in any way on this GWAS plot, and further, the significant SNPs in those other scaffolds aren't really outliers they are just not part of scaffold 11. Maybe rephrase?

-Supplement page 9: can you please clarify that the reason your bar plots show 1125 individuals while your Table S3 shows 1181 is because you have 56 without phenotype? (maybe the scat samples?)

-page 14: Clarify which table you mean by Table SX?

Very best wishes,
Anna Santure

Decision letter (RSPB-2021-1452.R0)

13-Jul-2021

Dear Mr Tietgen:

Your manuscript has now been peer reviewed and the review has been assessed by an Associate Editor. The reviewer's comments (not including confidential comments to the Editor) and the comments from the Associate Editor are included at the end of this email for your reference. As you will see, the reviewer and the Associate Editor are positive but have raised some issues that we would like you to address.

Research ethics:

Use of animals and field studies:

It is a condition of publication that you make available the data and research materials supporting the results in the article (<https://royalsociety.org/journals/authors/author-guidelines/#data>). Datasets should be deposited in an appropriate publicly available repository and details of the associated accession number, link or DOI to the datasets must be included in the Data Accessibility section of the article (<https://royalsociety.org/journals/ethics-policies/data-sharing-mining/>). Reference(s) to datasets should also be included in the reference list of the article with DOIs (where available).

Please submit a copy of your revised paper within three weeks. If we do not hear from you within this time your manuscript will be rejected. If you are unable to meet this deadline please let us know as soon as possible, as we may be able to grant a short extension.

Best wishes,
Professor Hans Heesterbeek
mailto:proceedingsb@royalsociety.org

Associate Editor
Comments to Author:

I thank the authors for submitting a revised manuscript that has been revised to address comments from two reviewers. This revised manuscript has been addressed by one reviewer, who was generally satisfied with the revisions and has made a few minor comments asking for additional revisions. I have a few additional minor comments of my own:

On line 411, it's not clear to me what "larger tendency" refers to. Perhaps "larger difference in fitness"?

Figure 1 is a bit hard to read and could use larger text in the axis labels and key. I also agree with the reviewer's comment about needing to adjust the figure to make the inclusion of additional chromosomes in the dog genome clearer.

Reviewer(s)' Comments to Author:

Referee: 2

Comments to the Author(s).

Thanks for the opportunity to review the revised version of this manuscript. The authors have done an excellent job of addressing mine and the other reviewer's comments and I appreciate the significant work that has gone into doing so. Below I list (i) a few quick comments in response to their cover letter / response to reviewers (reviewer comments that I haven't talked about below I think have been VERY well explained and resolved by the authors, so all I can say is thanks for all that work to do so!) and (ii) some very minor suggestions I came across on a final read through the manuscript:

(i) response to cover letter

-in their cover letter the authors acknowledge that a small error in calculating annual individual fitness meant that the p-value for female annual individual fitness was no longer significant. I appreciate the authors fixing this analysis (and changing text in the manuscript to acknowledge this) and agree with them that the change has not impacted the overall conclusions of the manuscript.

-Reviewer 1 comments

R1a: It would be good to make clear the reasoning for discarding the BLAST hits to chromosomes 27 and 17, and the evidence that the hits are all from the same locus.

-Although the authors explain their reasoning for discarding these hits in their response, I think this does need a quick mention somewhere in the manuscript files. Probably the best place for this is just a footnote to Supplementary Table S14 explaining that the SNPs also aligned to Chr 5 with slightly lower blast scores and so most likely do reside on Chr 5, but were conservatively discarded from further analysis (or something to that effect)

R1b: On that note, I find the Supplementary figure 3 quite convincing, and I think it could be part of Figure 1. I concede that this probably is a matter of personal taste.

-I liked the idea but not the implementation of the changes to Figure 1 in response to this comment. Scaffold 11 looks like it is rather two genes as per the top panels. Scaffold 2224 looks like a value of negative 2224 i.e. -2224. It isn't super clear which is Scaffold 68 vs 1772. How about putting 1772 on the same plane as 11 and 2224 on the same plane as 68, and putting the associated scaffold numbers consistently right and align-centred with the scaffold i.e.

```
11----- 1772-
                2224-                68-----
```

Reviewer 2 (i.e. my!) comments

R2a: long-winded question about population structure

-great response, thanks! And thanks for trying the RepeatABEL model, interesting that it found more significant SNPs! Please just add the lambda values also to the plots you now give in the supplement, the lambda values for the first two are in the main text, but not the final one.

-I'd love to also see the MDS plot coloured by origin. I agree that your GenABEL model has accounted for whatever population structure might be there, but it would be nice just to have an idea of the overall population structure across the range. Not essential, though!

R2b: other long-winded questions about understanding the data a bit better, the fitness models, me misunderstanding lambda and the GWAS

-nothing to add, just a thanks for your detailed responses and the addition of the clear tables and text, and congrats on the publication!

R2c: my comment re original line 88 was more to ask about whether the MC1R variants were in the extracellular vs the intracellular vs the transmembrane domain of this trans-membrane protein. This is interesting because it has implications for what protein-MC1R interactions might be most likely to be messed up. But, no problem if you're not sure, it is a pretty minor detail!

R2d: I asked about the p-value threshold in Figure S7 (now Figure S4) and you replied that the threshold should be at $-\log(1.24e-5)=4.9$.

-Apologies, I put the wrong number in my question! What I was querying here was the difference in the p-value threshold between the colour GWAS and the fitness GWAS but I realised from your response that the Bonferonni thresholds are of course different because there are different numbers of SNPs tested (genome vs region). Perhaps this could be clarified in the caption for what is now Figure S10?

(ii) minor suggestions

-line 299: suggest "The BLAST results also show that the four scaffolds that mapped to chromosome 5 and contain significant SNPs assemble next to each other (Figure 1a).

-line 413: add "do" to read "and thus do not represent"

-Supplement page 8: can you please clarify how many SNPs were kept / discarded when you say "The analysis was restrained to SNPs lying on arctic fox scaffold 11 and that matched with a position on dog chromosome 5 during the BLAST". I am also not sure about the statement "SNPs on other scaffolds or that matched with different dog chromosomes would naturally appear as outliers in Figure S2" - first, do you rather mean Figure S4 (the GWAS) and second I am not sure 'outliers' is the right word here as SNPs within scaffold 11 that matched with another dog chromosome would not be highlighted in any way on this GWAS plot, and further, the significant SNPs in those other scaffolds aren't really outliers they are just not part of scaffold 11. Maybe rephrase?

-Supplement page 9: can you please clarify that the reason your bar plots show 1125 individuals while your Table S3 shows 1181 is because you have 56 without phenotype? (maybe the scat samples?)

-page 14: Clarify which table you mean by Table SX?

Very best wishes,
Anna Santure

Author's Response to Decision Letter for (RSPB-2021-1452.R0)

See Appendix B.

Decision letter (RSPB-2021-1452.R1)

07-Sep-2021

Dear Mr Tietgen

I am pleased to inform you that your manuscript entitled "Fur colour in the arctic fox – genetic architecture and consequences for fitness" has been accepted for publication in Proceedings B.

Data Accessibility section

Open Access

Your article has been estimated as being 11 pages long. Our Production Office will be able to confirm the exact length at proof stage.

Paper charges

Sincerely,
Professor Hans Heesterbeek
Editor, Proceedings B
mailto: proceedingsb@royalsociety.org

Associate Editor:
Board Member
Comments to Author:
(There are no comments.)

Appendix A

Response letter RSPB-2020-1850

25th of June 2021

Dear Prof. Heesterbeek, dear Ms Singh-Shepherd, dear Board Member, dear reviewers,

Thank you for the very thorough and helpful reviews! We are glad you found the manuscript interesting to read and appreciate the opportunity to resubmit a revised version of the manuscript. We have improved the manuscript based on your comments and explained our choices in this response letter. Line numbers given in this letter correspond to line numbers in the “track changes” version of the revised manuscript.

If there still are some questions or comments concerning our study, we are of course happy to answer them!

During the revision process we found a small coding mistake in our R-script for the selection analysis (for the calculation of annual individual fitness lambda, the v1 values was based on a wrong age class). We adjusted this and re-ran the analysis for annual individual fitness. With the adjusted values, the p-values for female and male fitness increased and we want to note that the p-value for female annual individual fitness changed from 0.038 to 0.095. The genetic analyses as well as the analysis of the different fitness components were not affected by the error and remain as they were. We apologize for this! Generally, we think the broader picture of our study has not changed.

Lastly, we would like to ask that, in the case of acceptance, Øystein Flagstad is flagged as corresponding author.

Kind regards,

Lukas Tietgen and co-authors

Dear Mr Tietgen:

I am writing to inform you that your manuscript RSPB-2020-1850 entitled "Fur colour in the arctic fox – genetic architecture and consequences for fitness" has, in its current form, been rejected for publication in Proceedings B.

This action has been taken on the advice of referees, who have recommended that substantial revisions are necessary. With this in mind we would be happy to consider a resubmission, provided the comments of the referees are fully addressed. However please note that this is not a provisional acceptance.

1) A ‘response to referees’ document including details of how you have responded to the comments, and

the adjustments you have made.

- 2) A clean copy of the manuscript and one with 'tracked changes' indicating your 'response to referees' comments document.
- 3) Line numbers in your main document.
- 4) Data - please see our policies on data sharing to ensure that you are complying (<https://royalsociety.org/journals/authors/author-guidelines/#data>).

Sincerely,

Professor Hans Heesterbeek
mailto: proceedingsb@royalsociety.org
Associate Editor

Board Member: 1
Comments to Author:
Dear Mr. Tietgen,

Thank you for your submission to our special issue on Wild Quantitative Genetics in Proc R Soc B. Your manuscript has now been reviewed by two experts in the field. Both reviewers were positive about the manuscript and many of their concerns could be alleviated by rewriting. I do agree with reviewer 2's suggestions that reanalyzing the data with different controls for population subdivision could improve the paper.

Our response: Thank you for your comment. We are glad to here that the reviewers were positive about the manuscript and appreciate the valuable comments they had on the manuscript. Please see our response to reviewer 2 concerning population structure in the analysis.

Reviewer(s)' Comments to Author:

Referee: 1

Comments to the Author(s)

This paper presents genome-wide association of fur colour as a binary trait in Arctic foxes, and quantifies the fitness effects of this locus. The result confirms a previous candidate gene association at MC1R, and suggests that blue colour is associated with higher fitness.

The genomic methods for mapping are appropriate, and the paper is well-written. I have to declare that I don't have the expertise to evaluate the details of the demographic model of fitness.

I have a few comments about the use of Gene Ontology enrichment to analyse the potential for pleiotropy, the bioinformatics of mapping significant loci to chromosomes, and data availability.

Gene Ontology enrichment as potential for pleiotropy

The paper discusses the potential importance of pleiotropy and hypothesises about different causes for correlated effects (either other genes are in LD or dragged along by selection on MC1R (lines 399-422), or there are functional interactions with other melanocortin-regulated traits (lines 94-95)). However, the only evidence of pleiotropy is gene ontology enrichment, meaning that genes of certain annotations are close to MC1R in the dog genome. Thus the Gene Ontology analysis, in my opinion, needs to be treated much more cautiously.

Gene Ontology enrichments applied to regions of the genome are known to be subject to false positives (Pavlidis et al. 2012 <https://academic.oup.com/mbe/article/29/10/3237/1032149>). Furthermore, there is the added uncertainty of mapping genes between fox and dog, as well the uncertainty in the annotation (e.g., the first term listed in the supplementary table of enriched GO terms has been declared obsolete by the GOA).

Our response:

We agree with the reviewer that GO enrichment analyses have the potential problem of false positives. Furthermore, the GO terms are usually so general that they unfortunately do not give very much useful information on the specific functions of putative loci related to the trait(s) of interest. These potential problems were among the reasons why we in addition to the GO enrichment analysis examined the putative functions of genes physically very close to (and in strong LD with) the significant SNPs in our GWAS. These putative functions were determined from primary literature and were therefore not based on the more general GO terms. Nevertheless, we appreciate the problematic possibility of “storytelling” (Pavlidis et al. 2012) and have therefore made some small adjustments to the text (Lines 409-411; 499-501) to tone down the GO enrichment analyses and show the uncertainty in the putative effects and potential importance of linked genes in arctic fox.

As pointed out by the reviewer, there could be some uncertainty regarding the annotation of genes in Arctic fox genome vs the dog genome. Chromosome painting suggests that there is high synteny

for chromosome 5 (on which MC1R is located) between dog and arctic fox (Graphodatsky et al., 2000 <https://doi.org/10.1023/A:1009217400140>). Consequently, we believe that there is little error in our suggested putatively important genes linked to MC1R. The explanation of this has now also been improved in the revised manuscript (Lines 279-280).

You also mention the GO term that has been coined obsolete. This status is due to some reorganising in the GO terms and has, as far as we understand, nothing to do with differences between the dog and arctic fox genomes (history of the GO term: <https://www.ebi.ac.uk/QuickGO/GTerm?id=GO:0044424#term=history>)

Matches to more than one chromosome

The BLAST search and Manhattan plot (supplementary figure 3) show significant associations to more than one chromosome. It would be good to make clear the reasoning for discarding the BLAST hits to chromosomes 27 and 17, and the evidence that the hits are all from the same locus.

On that note, I find the Supplementary figure 3 quite convincing, and I think it could be part of Figure 1. I concede that this probably is a matter of personal taste.

Our response:

We agree with the reviewer that how we presented the locations of significant GWAS hits on different scaffolds and chromosomes were a bit unclear. This has now been improved by including information on which dog chromosome the hits on different scaffolds blasted to in Table S10, and by including scaffold locations in Figure 1, which now clearly shows that 486 out of the 489 significant hits that blasted to the dog genome actually are located in this ca. 25 Mbp region on chromosome 5. We do however wish to keep the Manhattan plot showing all scaffolds in the supplementary material as Fig S2.

Regarding the two hits on chromosome 27, both these SNPs had many BLAST hits that met the requirements we set (e-value < 0.001 and query coverage > 70% [50 bp]) and among these there were actually also hits on chromosome 5. We originally decided to choose the SNPs with the lowest e-value to have an objective filtering method and ended up listing the three hits on chromosome 17 and 27. Due to this procedure, and the sheer number of hits on chromosome 5 we believe that the hits on chromosome 17 and 27 are artefacts of the BLAST algorithms and did not consider them further.

Accuracy of fitness measurement

The supplementary analysis with known pedigree suggests that parentage assignment is accurate. However, I am left wondering how certain we can be that the pedigree (from the wild animals) is complete i.e. how accurate is the estimate of the number of pups? Is this obvious from the quality of the monitoring data, or is there room for uncertainty? As far as I can tell, this is not addressed in the paper.

Our response:

As noted by the reviewer, since we are working with a wild population, there will be some uncertainty regarding the number of pups produced by a given individual and survival data.

Extensive monitoring over a large time period including genetic sampling has ensured that the monitoring programme has extremely good knowledge about the distribution of the arctic fox in Norway. It is thus possible to detect virtually all breedings. However, there will always be the chance that some breedings or dens remain unknown. The biggest uncertainty lies within the measure of number of pups an adult individual produces due to the high mortality rates during the first winter for juveniles. Our fecundity measures use the number of recruits instead of pups, thus avoiding the high uncertainty there. Also for the measure of breeding probability, the number of pups is not important and as long as one pup recruits and is detected, the breeding is detected.

Also, there is no reason to believe that missing breedings or observations of recruits would introduce any bias to our analyses of effects of colour morph genotypes on fitness, because such uncertainties are randomly distributed across genotypes within each subpopulation.

An explanation was added in the methods (lines 202-209)

Data and code availability

The reviewer instructions specify that we must check all supplementary material to ensure that all data has been submitted. It appears that the underlying data was not available for review.

The paper states that data and scripts will be made available on Dryad. It would have been useful to see these as a reviewer, especially since the verbal descriptions of the models (lines 193-200) are not easy to follow, even with the aid of the Supplementary materials.

Our response: Data will be made available on Dryad (<https://doi.org/10.5061/dryad.nk98sf7th>).

Minor comments

Introduction, lines 68-72: This passage seems important for the framing of the study, but it is hard to follow exactly what it says and why it "demonstrates the importance of more studies ... into the genetic architecture of adaptive traits". Is the point that multivariate selection on correlated traits complicates prediction of the selection response? How does this study contribute to illuminating this question?

Our response: We agree that this part was a bit unclear and have adjusted the text to make our message clearer.

Lines 93-94: I agree that candidate gene approaches have serious pitfalls. However, it also seems to have worked reasonably well for pigmentation in multiple species, including this one; this study finds, indeed that the expected candidate gene is a major locus for the trait.

Our response: It is correct that the candidate gene approach has worked well in a number of cases (despite potentially serious problems), but we wish to emphasize the importance of independently verifying even apparently successful candidate gene analyses using a robust GWAS approach, as we do in the current study. Note that these issues have also been mentioned in the Discussion (lines 424-428; 499-503).

Methods, lines 117-119: Some more detail about the scat sampling would be good. Did those data include fur colour phenotypes?

Our response: Some more information on the scat sampling, including a couple of references, has been added to the Methods (lines 130-138). Fur colour was only registered when a fox was observed or handled. The selection analyses which to a degree are based on scat sampled data, only used “fur colour genotype” (i.e. the genotype of the most associated with fur colour).

Figure 1b and 2: Wouldn't confidence intervals for the difference in fitness be more useful than 95% prediction intervals?

Our response:

We fully understand the desire to see the differences in fitness directly. Still, we want the figures to show the predictions for each genotype as this is informative for the mean individual fitness of each genotype. The differences and CI you request was already available in the supplementary material (Tables S8-S11), but we have now added a sentence in the table captions to explain the parameters and have also added the estimated differences directly in the main text of the results. We hope that you agree that this is a good solution to your request.

Referee: 2

Comments to the Author(s)

Thank you for the opportunity to review this paper, it has been a pleasure to read and to work through the analyses. It has been particularly impressive to see such a comprehensive treatment of the fitness consequences of the diagnostic SNP for the presumed MC1R ‘causative’ haplotype, and I enjoyed the discussion of the possibility of effects of other nearby genes via linkage and pleiotropy. I also think it is great that you’ve used a robust statistical approach to test for association to be able to confirm previous analyses. The writing is clear and analyses are generally well explained (although my apologies, I was very confused by the fitness analyses!). For clarity I think it would have been nice to include some more of the supplementary information in the manuscript, but I appreciate that the page limits are rather restrictive which does unfortunately force brevity in the methods and results sections. I have a few general comments and questions and some smaller edits. Please note that despite the length of my comments I do think this is an excellent manuscript and I commend you on some fantastic science!

Major comment 2.1

The first general comment is around the choice to merge datasets across the range to test for association. I appreciate that this was done to increase sample size to detect power for association, but I am a little worried at the high genomic inflation (you do not provide a Q-Q plot to show the distribution of p-values – please do so in your supplement -, but they clearly seem to be skewed by more than simply your

significant peaks). From my understanding your GWAS model includes only the GRM as a random effect and there are no other fixed or random effects fitted? This is possible to do in RepeatABEL (even if you don't have repeat measures) and we found Lars Ronnegard incredibly helpful when we fitted our hihi GWAS (<https://doi.org/10.1098/rspb.2020.0948> - the dryad link for this paper has R code for fitting a model with multiple random and fixed effects in RepeatABEL) or I can also offer some pointers! I'd recommend using a structure-like analysis to test whether you have very clear structure in your dataset across subpopulations, and include subpopulation as a fixed effect if so, and then also include the GRM to account for fine-scale within population structure. There is a loss of power from 'double fitting' this population structure but the GWAS peak is so clear I'd think the most likely outcome is that your three additional peaks might disappear, which may not be a bad thing... my feeling is that these are either misassembled scaffolds that should be part of 11, or are picking up some level of structure in your data that hasn't been adequately accounted for. I was also not sure what effect the inclusion of the breeding population had on your overall structure. It seems this is a very mixed population and that animals are captive bred and then released to many of the subpopulations, so some populations are more 'pure' than others and some might be quite mixed. Perhaps the 'origin' should also then be added as a fixed effect to your GWAS?

Our response:

We thank the reviewer for very thorough and helpful comments on our manuscript. You are correct that populations were merged to increase statistical power. To explore the potential bias in our results by doing so we have followed your suggestions. First, we added the Q-Q plot to the supplementary materials (Figure S2a). We also added a Q-Q plot after having removed the four scaffolds with significant SNPs (Figure S2c). The second Q-Q plot is, as expected, much less skewed. We interpret this that the strong skew in the original Q-Q plot is mostly due to the unusually large number of highly significant SNPs.

You are also correct that the GWAS in our analysis only includes the GRM without any other fixed or random effects. Hence, we reinvestigated our analysis based on your comments. First, we performed the GWAS including the three first principal components (PCs) obtained through classical multidimensional scaling (MDS). The lambda estimate (inflation factor in the GWAS) was virtually unchanged compared to the analysis in the manuscript (1.902 vs. 1.917). The corresponding Q-Q plot also seems to be unchanged (Figure S2b). The three first PCs explained approximately 6%, 5% and 3.5% of the total variation in the data. A cluster plot of the first two PCs with the colour morph of each individual did not reveal any strong structure concerning arctic fox fur colour in the data set (Figure S3). We now explain this in the methods section (line 158-164).

We also tried to fit the GWAS in a RepeatABEL framework as suggested by the reviewer. We followed the RepeatABEL documentation, and the R-script kindly provided by you. We tried to fit the model with only *subpopulation* as a fixed effect, as well as *subpopulation* plus *origin*. Including both fixed effects led to an increase in significant SNPs to 950. These were distributed over the same four scaffolds as before (11, 68, 1772, 2224). For unknown reasons, we were not able to extract an estimate for lambda from this model, as the output was given as NA.

Based on these additional analyses we believe that our results from the original analyses in the manuscript are robust and not the result of e.g. genetic structure. We thus decided to keep the analysis as it is in the manuscript.

The arctic fox genome that we used is not a completely annotated genome and the scaffold assembly may have some minor faults. All four scaffolds with significant SNPs BLAST to dog chromosome 5.

We have added the scaffolds into Figure 1 to make this point clear. What we can see there is that scaffold 68 is the natural continuation of scaffold 11. The scaffolds in the assembly are ordered by length, meaning scaffold 0 is the largest (in terms of bp) and scaffolds with high numbers such as 1772 and 2224 are extremely small scaffolds. These two specific scaffolds have four and two SNPs respectively, of which three and two are significant (see Table S14). We have also updated Table S10 with information on which dog chromosomes the different scaffolds BLAST to. We have added references to the new Figure 1a and Table S14 several places in the manuscript and added some lines of further explanation (line 300-301; 412-414)

Major comment 2.2

The second general comment is to do with understanding the data a bit better, I think it is a pity to not have Figure S1 (or some version of) in the main text, this would be particularly helpful if white/blue proportions were represented for each subpopulation on the map. See my comment below with regard to supplementary table S1, it would have been really great to have the categories of colour, location, origin, sex broken down more as I found it hard to assess the evidence that you'd used to decide that neither camouflage or thermoregulation were likely selection pressures for colour morph. I appreciate that this information is likely more thoroughly explored in the paper that is in review, but I think a little more information is needed to put the analyses / conclusions of this manuscript in context.

Our response:

We agree it is useful to have more information about the data structure and have added pie charts showing the proportions of white and blue foxes in each subpopulation across all years to Figure S1. We agree that it would be nice to have this figure in the main text, however, it seems as if we are not able to do so within the page limits.

We also think you have a good point regarding the information given in Table S1. To make in the inclusion of additional information feasible and keep the table readable, we have opted to include three different tables in the supplementary instead of the old Table S1. We followed your suggestion from further down in this document and organised the tables according to latitude.

Table S1 gives sample sizes of the genotyped individuals and the genetic analyses and remains basically the same as before as we feel that this is sufficient for this part of the study. Table S2 presents numbers of released fox individuals.

Table S3 is an enhanced version of the previous Table S1 and presents more detailed sample sizes for the fitness analyses. Part a) is based on individuals whereas part b) is based on annual observations (i.e. one fox being observed for three years results in three annual observations).

The paper that explores more in detail how environmental effects (i.e. snow cover and winter temperature) interact with colour morph phenotype to affect fitness components has now been published (Di Bernardi et al., 2021 <https://doi.org/10.1111/1365-2656.13457>). In addition, we have adjusted the part of the Discussion that focus on this and more details are now given in this manuscript as well (lines 447-458).

Major comment 2.3

The third comment is around the statistical models fitted to test overall fitness and components thereof and how to understand what you term predicted values in Figure 1 and 2. I know you have summary tables and extensive text in the supplement but I am quite confused by what effects were included in which models and exactly how many models were fitted. For example in terms of effects on the six measures of fitness (male fitness, female fitness, fecundity, survival, etc), the following terms are mentioned: rodent and snow phases as shown in Tables S3 and S8, genotype and sex (Tables S4-S7), age, age², origin (Table S8), interactions thereof (Table S8), subpopulation and year (Figure S6). Mentioned in the main text but nowhere in the supplement are random intercepts for birth year and den. Were all of these terms fitted together into a single model for each of the six fitness phenotypes? Or were models for male and female fitness different from models for the other four fitness phenotypes? Or perhaps two models per phenotype as you test models with and without interactions (Table S8). Or did each fitness phenotype have a series of models that independently tested environmental models from the models that included other effects such as sex? I think all of this would immediately be cleared up by a summary table that lists every single model fitted with the fixed and random effects, has sample sizes for the number of individuals and number of phenotypes that makes clear what are repeated measures, and perhaps has a column that shows where the results of this model can be found in terms of supplementary tables and figures.

Our response: We acknowledge that some of our descriptions around the modelling in the selection analysis has been confusing. We have thus made a range of adjustments.

Basically, we have a measure for annual individual fitness (λ). As this is calculated for each sex independently, we have separate models for male and female foxes. Additionally, we model four different fitness components to decompose the overall fitness. These are *fecundity* (i.e. no recruits produced), *adult annual survival*, *adult breeding probability* and *recruitment probability* (i.e. juvenile survival). In total this sums up to six different response variables (λ (females), λ (males), four fitness components). We have added Table S6 to give an overview about the models and the variables included in those to make this more understandable.

In terms of environmental variables, we were interested in whether the genotype effect on individual fitness or the fitness components is affected by the different environmental variables (i.e. whether there is an interaction between genotype and the environmental variable). Therefore we have fitted one model including the environmental variable as an additive term and one model with the interaction term *genotype x environmental variable*. These were compared using LRT. In cases where sex or age were previously found to affect the response variable (should now be clear from Table S6), these were included both in the additive and interaction model.

Further we have made some minor adjustments here and there to improve coherency. We have added sub headlines in the *selection analysis* section in the methods to differentiate better between individual fitness models, fitness component models and environmental variables models. We have also written response variables in *italics* when named for the first time in the methods section to highlight these. We have added “fitness components” before “models” (line 239) to make clearer what models we are talking about. We have specified what “adult fitness components” are (line 240-241). We have improved our explanation of testing environmental variables (lines 255-261). We added estimates for random intercepts in the model summary tables in the supplementary.

Major comment 2.4

A fourth point where I am confused is what is meant by lambda. I would think of this as a whole lifetime measure of reproductive success i.e. a single value per individual. You mention predicted lambda and I therefore interpreted this as a single standardised fitness measure for each individual, standardised for all the significant effects of origin, subpopulation etc. However, in the description of the fitness GWAS in the supplement you use RepeatABEL to be able to use repeat measures of lambda. What are repeat measures of lambda? Why the need for RepeatABEL? And why can't you use the predicted lambda from your selection analyses to run a GWAS on? Further, why isn't origin included in the fitness GWAS? Again, some of this might be cleared up by a simple table that, similar to your fitness table, lists clearly the GWAS models, what the phenotypes are, and where to find the results. I did wonder if you used a single predicted lambda for your phenotype in the GWAS rather than what seems to be annual breeding data (???) whether your pvalue landscape might have looked a bit different? I would think it would have been quite nice to do this genome-wide. This of course makes the assumption that fitness has a significant heritable component, which is appearing rather unlikely if the diagnostic colour SNP is nowhere near significant in this GWAS, despite the relationship between genotype and fitness. The hint at heterozygote advantage at the diagnostic SNP may suggest that the fitness GWAS should be fitted as a dominant model, but these are drastically underpowered, and I don't think would perform very well given your lack of TT individuals.

Our response:

We understand that the explanations regarding lambda as a measure for individual fitness were a bit too brief in the manuscript. We have tried to clarify the meaning of lambda in the methods (lines 211-218). Lambda is an age-independent measure of annual individual fitness. Thus, this is why we can have repeated measures of individual fitness for individuals in our analyses. Hopefully, this makes the analyses much easier to understand and resolves most of the questions that you had regarding analyses involving individual fitness.

You have a valid point about including *origin* in the fitness GWAS and we have now done so. The analysis is also updated with the new fitness measures. Otherwise, we wanted to keep this supplementary investigative analysis as linked as possible to the genetics, thus restraining it to the given area of the genome. We also agree that a dominant model might be appropriate but not feasible with the lack of data on TT foxes.

Major comment 2.5

Fifth, is this really a single gene trait? What has happened with the 'incorrect' phenotypes where genotype does not match morph? Incorrect phenotyping or sample mixups perhaps? I would be surprised if it is just imperfect LD with the causative SNPs, but perhaps that is an option? Are these morphs intermediate in any way that might suggest a modifier locus? Are they only one sex that might even suggest a sex-specific modifier? Assuming they are not some artefact of the GWAS not accounting for population structure adequately, are they true different genomic location from the main peak? Is there any sign that the additional scaffolds could be placed on dog chromosome 5? What about blast matches with non-significant nearby SNPs, even if the top SNPs are not blasting to chromosome 5, that might suggest

these smaller scaffolds should be added to / inserted into scaffold 11? I think you need to discuss the phenotype mismatch options in light of the three other significant peaks in your GWAS.

Our response: We strongly believe that this is a single gene trait. First, the cases in which the recorded phenotype does not match the expected genotype are very few (we have added Figure S6 and Table S4 to the supplementary to make the data even clearer). We assume this to be due to incorrect phenotyping in the field or mistakes in data management afterwards. Field data is collected by a wide range of people under sometimes hard field conditions, which in our opinion is a reasonable explanation of these mistakes (added sentence in discussion, line 420-422). Second, there are no intermediate morphs, nor did we see any bias in sex among the individuals where genotype and phenotype did not match. There is a very rare third colour morph called the “sandy morph” for which we lack data. We added information regarding the third morph (line 78-79).

As explained in the responses to comments of reviewer 1 and your major comment 1 above, all scaffolds with significant SNPs actually blast to the same ca. 25Mbp region of dog chromosome 5 which now is shown in the edited Figure 1 and Table S14.

Minor comments / edits

Main text

-line 58: I think you could be a bit more assertive here! “Colouration is one of the most conspicuous...”

Our response: We agree and changed the sentence accordingly.

-line 68-72: this seems to be written in a way that suggests that there isn't theory yet to make these predictions, but of course there is. I think the point you are (correctly) trying to make is that measuring traits, their relationship to fitness, and their correlations to each other, is challenging in wild populations. Perhaps rewrite a little to make this a bit clearer?

Our response: We changed our wording to highlight the challenge of obtaining such data in wild populations.

-line 74: could you say and describe exactly how many morphs, rather than ‘multiple’? It made me wonder if some are perhaps not quite so distinct from each other? That could represent a simple genetic basis for white vs blue but some nuance in gradients between these morphs determined by other loci (see question above about the phenotype ‘misclassifications’ and additional scaffolds above)

Our response: We added a sentence about the rare third colour morph (line 79). This morph is very rare and we lack (genetic) data on this morph.

-line 79: delete ‘the’ to read ‘is the result of dominant allele effects’ or perhaps ‘is a result of the effect of a dominant allele’

Our response: Changed according to your second suggestion.

-line 88: it felt like a descriptor was missing here e.g. extracellular / intracellular to read e.g. ‘within the intracellular region of the...’

Our response: Added descriptor “intragenic” to specify location of the substitutions.

-line 94: is ‘gene complex’ the correct phrase here? I think of a gene complex as being interacting genes in a protein complex e.g. to make a heterodimer. I feel like ‘gene family’ might be what you meant, as I am pretty sure MC1R does not form a complex directly with any of these other MCxR proteins?

Our response: We agree that “gene family” is the correct definition here and changed accordingly.

-line 112: is it correct, as indicated by line 115 “the breeding station” that there is only one captive facility? It would be good if this could be stated clearly. Also, I think it is important to briefly state that “breeding pairs in the breeding station represent all extant subpopulations in Scandinavia and are chosen to maintain genetic diversity” which is currently on line 430-431 but I think would be helpful here in the methods section instead / as well as in the discussion.

Our response: Yes, there is only one breeding station. We added a sentence to clarify this (line 118-121)

-line 117-119: I was not really sure why these foxes are mentioned? Since they have no phenotype and no fitness? Or do you mean they were included in some of the fitness calculations, just not the juvenile survival, because you could put them in the pedigree and hence calculate a fecundity measure for them for a given year?

Our response: To clarify things here, we decided to only say how many foxes were identified through scat samples during winter monitoring in line 127. We cannot be certain about the birth year of these foxes and thus cannot include them in the analysis for juvenile (first year) recruitment probability (survival). We explain this in a more suitable paragraph (line 235-238)

-line 123: can you please confirm that only arctic fox specific SNPs were analysed? There are none that are polymorphic between arctic fox and red fox? Or if there are, these were excluded from these analyses?

Our response: Yes, we only used SNPs identified as polymorphic in our sample of ca. 700 genotyped arctic foxes in our analyses. This has now been pointed out in the main text (line 143).

-line 165: I didn’t get a feeling from this about recapture rates. For example, how often do you miss finding a fox in a particular census year but locate it again the following year?

Our response: We are confident that we have high recapture rates due to the extensive monitoring effort that also included genetic sampling. 10% of our study individuals were not seen in a given census but appeared later on. Even if we do not have a perfect sampling of individuals each year this should not bias out results. The analyses only requires a sample of individuals each year and if some individuals are missed because we did not see them this missing data is very likely of a random type. Such that there is no association between not being recorded and the measurement that we make. A short statement regarding this is added to the methods (line 200-207)

-line 169: I also didn't get a feeling for what proportion of individuals in the wild would have their breeding den located and pups genotyped? Line 173-174 you assume these individuals to have not bred, but how realistic is that? What impact might missing data have on the wild population?

Our response: This is answered in our response to reviewer 1.

-line 202: add 'a' to read 'is to a large extent'

Our response: "a" added.

-line 209: I was a bit confused by this section as I think of heterozygote advantage as being at a single locus. I think this would be less confusing if this was instead termed as heterozygosity advantage / inbreeding depression as 'heterozygosity' is the term used when talking about heterozygosity-fitness correlations.

Our response: We agree with the reviewer that our use of the term "heterozygote advantage" was imprecise and have now instead used the term "heterozygosity advantage" as suggested.

-line 234: I was confused when I came to Figure S8 in the supplement as I wasn't sure where it came from. Can you add here that presumably most of this co-expression is from human / model organism data? Is it mammals only (or could be restricted to mammals only)?

Our response: Indeed, the analysis is based on the GeneMANIA's database for humans as no canines are available. Added a sentence to clarify this (line 290-291).

-line 243-244: I found it very interesting that there were only three SNPs that aligned elsewhere than dog chromosome 5 in the genome. I would like to know which dog chromosome the scaffolds 68, 1772 and 2224 aligned to (this could be easily done by blasting all the SNP flanking sequences from these scaffolds on to the dog genome). My guess is that some of these scaffolds might nest inside scaffold 11. If they don't, I really query whether you can say that MC1R is the 'sole causal gene' (see comments above). The peak on scaffold 68 looks particularly well supported by 13 SNPs and it would be nice to also zoom in to the genes through this region. The three and two SNPs on the other two scaffolds could possibly be misplaced, have you checked their second highest blast hit even if their first hit is not to dog chromosome 5? Or the blast hits of other nearby SNPs in the same scaffold that are not significant?

Our response: See our response to reviewer 1. Information added to Fig 1a and Table S14 should clarify the positions of the different scaffolds.

-line 346: I am not sure at the choice of the word 'covary'. I think you've taken this a shortcut to say that SNPs in these genes are in strong LD with the diagnostic SNP. This is not quite the same thing as the causative colour SNP being in LD with causative SNPs in other genes, which is what I think you imply by the term covary. I'd recommend a softening.

Our response: Changed the sentence to address your comment (line 407-409).

-line 355-356: see above, I am a bit unsure of the assertion that no other genes explain variation when (if) you have significant GWAS peaks on 3 other scaffolds

Our response: See our response above and to reviewer 1. The peaks on different scaffolds blast to the same dog chromosome (Figure 1a, Table S14).

-line 365: delete 'both' as you list three phenotypes

Our response: Deleted "both".

-line 373: although results are explained, it would have been good to see this uploaded as supporting information to assess these conclusions

Our response: We have changed the wording slightly and added some more details from the results of Di Bernardi et al., 2021 (lines 448-458). The paper is now published in JAE and all details can be found there: <https://doi.org/10.1111/1365-2656.13457>.

-line 377: I am not sure I see duration of snow fall tested in Table S8, you have first and last snowfall but not the difference between them. Have you tested for the effects of length of winter in your data?

Our response: This is a good point! As you say, we first and last snowfall but not the difference and it is true that we don't have any other measure for length of winter. The Di Bernardi et al., 2021 paper looks more detailed into these snow variables, while this paper only uses the two mentioned variables as proxies for winter conditions. We have changed the wording to make this clearer (lines 256-257).

Supplementary material

-Table S1: I appreciate the summary table is going to get very complicated to do this, but I would like to see a: split for sex, phenotype and origin (wild/captive) for each of the numbers in this table. I would also like to see the table ordered in the same order (i.e. by latitude) as in the map. This would allow readers to see (i) any systematic pattern in colour for location, (ii) any bias in male / female ratio for location, (iii) how the wild/captive breaks down for each morph. In addition, it would be good to have each sample listed with all of their accompanying information. I recognise that burrow sites are sensitive but surely they could all be grouped by subpopulation?

Our response: This comment is addressed within our response to your *Major comment 3*.

-Figure S3 and Supplementary Table 8: I really would like to see the mappings of your significant SNPs to positions on the dog genome. Please can you add columns with dog chromosome and position and arctic fox position to supplementary table 8. I really wondered where the scaffolds 68, 1772 and 2224 matched to in the dog genome, and further whether the 'gap' you see in Figure S3 is just a region of non-significant SNPs or whether this is a region of your assembly where there might be evidence of a clear

genome rearrangement event between dog and fox (which incidentally might also be caused by a missassembly of your scaffold).

Our response: Regarding the position of the different scaffolds, see earlier answers. We have added the columns you suggested to the Excel file in Supplementary material 8 and hope that this data will make the connections between the scaffolds more clearly.

It is possible that the region of non-significant SNPs represent an area of the scaffold that is misassembled. Improvements are frequently made to published genomes, such as the human genome, which has taken 20 years to complete. However, should this region of scaffold 11 be misassembled, it will not affect the results we present in this study.

-section on parentage analysis, second-to-last sentence: could you please explain what you mean by correct dummy parents being assigned even if parents are ungenotyped? Do you mean perhaps that sibs cluster together?

Our response: Yes, we mean that sibs cluster together with the same dummy parent(s). We now explain this in the sentence you referred to.

-supplementary material 9 – fitness GWAS: you say that the GRM was based on all SNPs that passed QC, in the main text you state these were LD pruned, which is correct?

Our response: This was a mistake in the supplementary. The GRM was based on the LD pruned dataset. This is now corrected.

-Figure S7 vs statement in main text lines 308-311 “Bonferroni-corrected significance level ($p = 1.24E-05$)”. The p-value threshold does not match between the graph and the main text and I cannot see any sign of a SNP ‘close to significant’ (line 310) on this plot. Please check.

Our response: The p-value in Fig S7 is given on a negative log scale: $-\log(1.24e-5)=4.9$, which is shown in S7.

-Supplementary material 11 – LD decay: I would have liked the dataset to have been pruned to remove close relatives before calculating LD (although I see the argument to keep them in as the GWAS is leveraging both family linkage and population LD to detect association). There are likely also population specific LD decays. I think you should just acknowledge clearly that all individuals were used and that this will include close family members and is across populations so readers have this clear in their mind.

Our response: Thank you for bringing this to the light. We have added a sentence in Supplementary material 11 to highlight the fact that individuals included came from different subpopulations and could be close relatives.

Very best wishes,
Anna Santure

Appendix B

Response letter RSPB-2021-1452

Dear Prof. Heesterbeek, dear Ms Singh-Shepherd, dear Ms Santure,

We are grateful for the opportunity to revise our manuscript and are glad that you valued our comments and revisions so far!

We have responded to the comments made in this second round of revision and believe the manuscript has been further improved. Read our responses further down in this document. Line numbers refer to the track changes document.

While going through the manuscript and double checking the data presented, we found a mix up with some numbers in the results section. The correct numbers are now given (lines 336, 342-343). We emphasize that no analyses were changed and thus no results are affected by this mix up.

The datasets and corresponding R scripts are uploaded to Dryad but not yet published. For the time being, the repository can be accessed through this temporary link:

<https://datadryad.org/stash/share/2oVqUIwvX2Qas2kg-QV17e7vUA98y1D7cPMcXvVIdZU>. Since the arctic fox is an endangered species in the study area, sensitive data (i.e., den locations) will not be released. Further, we would like to ask for an embargo due to the sensitive nature of the data. Upon publishing of the dataset, it can be accessed under doi:10.5061/dryad.ht76hrgd.

Lastly, we would like to ask for a small favour. Due to the scope of our study and the Proceedings page limit, we had to move some figures to the ESM. We think we are quite close to the 10 page limit with the manuscript now, however, are not entirely sure how much “space” there is. Is there any possibility to get information about whether or not there is some space left when the layout and formatting is underway? In this case, we would like to move Figure S9 into the main text. We have full understanding if this is not possible and are happy with the manuscript as it stands now.

Kind regards,

Lukas Tietgen and co-authors

Dear Mr Tietgen:

Your manuscript has now been peer reviewed and the review has been assessed by an Associate Editor. The reviewer's comments (not including confidential comments to the Editor) and the comments from the Associate Editor are included at the end of this email for your reference. As you will see, the reviewer and the Associate Editor are positive but have raised some issues that we would like you to address.

Research ethics:

Use of animals and field studies:

It is a condition of publication that you make available the data and research materials supporting the results in the article (<https://royalsociety.org/journals/authors/author-guidelines/#data>). Datasets should be deposited in an appropriate publicly available repository and details of the associated accession number, link or DOI to the datasets must be included in the Data Accessibility section of the article (<https://royalsociety.org/journals/ethics-policies/data-sharing-mining/>). Reference(s) to datasets should also be included in the reference list of the article with DOIs (where available).

Please submit a copy of your revised paper within three weeks. If we do not hear from you within this time your manuscript will be rejected. If you are unable to meet this deadline please let us know as soon as possible, as we may be able to grant a short extension.

Best wishes,

Professor Hans Heesterbeek
mailto:proceedingsb@royalsociety.org

Associate Editor

Comments to Author:

I thank the authors for submitting a revised manuscript that has been revised to address comments from two reviewers. This revised manuscript has been addressed by one reviewer, who was generally satisfied with the revisions and has made a few minor comments asking for additional revisions. I have a few additional minor comments of my own:

On line 411, it's not clear to me what "larger tendency" refers to. Perhaps "larger difference in fitness"?

Our response: We agree that the wording was unprecise and changed it according to your suggestion which seems more appropriate.

Figure 1 is a bit hard to read and could use larger text in the axis labels and key. I also agree with the reviewer's comment about needing to adjust the figure to make the inclusion of additional chromosomes in the dog genome clearer.

Our response: Font sizes have been increased (see also response to reviewer below).

Reviewer(s)' Comments to Author:

Referee: 2

Comments to the Author(s).

Thanks for the opportunity to review the revised version of this manuscript. The authors have done an excellent job of addressing mine and the other reviewer's comments and I appreciate the significant work that has gone into doing so. Below I list (i) a few quick comments in response to their cover letter / response to reviewers (reviewer comments that I haven't talked about below I think have been VERY well explained and resolved by the authors, so all I can say is thanks for all that work to do so!) and (ii) some very minor suggestions I came across on a final read through the manuscript:

(i) response to cover letter

-in their cover letter the authors acknowledge that a small error in calculating annual individual fitness meant that the p-value for female annual individual fitness was no longer significant. I appreciate the authors fixing this analysis (and changing text in the manuscript to acknowledge this) and agree with them that the change has not impacted the overall conclusions of the manuscript.

-Reviewer 1 comments

R1a: It would be good to make clear the reasoning for discarding the BLAST hits to chromosomes 27 and 17, and the evidence that the hits are all from the same locus.

-Although the authors explain their reasoning for discarding these hits in their response, I think this does need a quick mention somewhere in the manuscript files. Probably the best place for this is just a footnote to Supplementary Table S14 explaining that the SNPs also aligned to Chr 5 with slightly lower blast scores and so most likely do reside on Chr 5, but were conservatively discarded from further analysis (or something to that effect)

Our response: Good point! We followed your advice and included a footnote to Table S14.

R1b: On that note, I find the Supplementary figure 3 quite convincing, and I think it could be part of Figure 1. I concede that this probably is a matter of personal taste.

-I liked the idea but not the implementation of the changes to Figure 1 in response to this comment. Scaffold 11 looks like it is rather two genes as per the top panels. Scaffold 2224 looks like a value of negative 2224 i.e. -2224. It isn't super clear which is Scaffold 68 vs 1772. How about putting 1772 on the same plane as 11 and 2224 on the same plane as 68, and putting the associated scaffold numbers consistently right and align-centred with the scaffold i.e.

11----- 1772-

Our response: We agree that the implementation left some room for improvements. There is a lot going on in the figure already, so we acknowledge that adding further elements might make it more confusing. We have tried to make the implementation of the scaffold information clearer by giving it some more space and have the scaffold numbers consistently on the left side of the according line. Font sizes have also been increased a little (ref comment from Associate editor).

Reviewer 2 (i.e. my!) comments

R2a: long-winded question about population structure

-great response, thanks! And thanks for trying the RepeatABEL model, interesting that it found more significant SNPs! Please just add the lambda values also to the plots you now give in the supplement, the lambda values for the first two are in the main text, but not the final one.

Our response: We have added the lambda values in the caption to Fig S2, and added a sentence about Fig S2c in the main text (lines 156-159).

-I'd love to also see the MDS plot coloured by origin. I agree that your GenABEL model has accounted for whatever population structure might be there, but it would be nice just to have an idea of the overall population structure across the range. Not essential, though!

Our response: The plot is now added as Figure S3b.

R2b: other long-winded questions about understanding the data a bit better, the fitness models, me misunderstanding lambda and the GWAS

-nothing to add, just a thanks for your detailed responses and the addition of the clear tables and text, and

congrats on the publication!

R2c: my comment re original line 88 was more to ask about whether the MC1R variants were in the extracellular vs the intracellular vs the transmembrane domain of this trans-membrane protein. This is interesting because it has implications for what protein-MC1R interactions might be most likely to be messed up. But, no problem if you're not sure, it is a pretty minor detail!

Our response: Although interesting, we chose not to look into this issue as the intracellular functions of specific proteins is beyond our expertise.

R2d: I asked about the p-value threshold in Figure S7 (now Figure S4) and you replied that the threshold should be at $-\log(1.24e-5)=4.9$.

-Apologies, I put the wrong number in my question! What I was querying here was the difference in the p-value threshold between the colour GWAS and the fitness GWAS but I realised from your response that the Bonferonni thresholds are of course different because there are different numbers of SNPs tested (genome vs region). Perhaps this could be clarified in the caption for what is now Figure S10?

Our response: Yes, you are right about the different number of SNPs affecting the Bonferroni-corrected thresholds. We have added a clarifying sentence in the caption of Fig S10 as suggested.

(ii) minor suggestions

-line 299: suggest "The BLAST results also show that the four scaffolds that mapped to chromosome 5 and contain significant SNPs assemble next to each other (Figure 1a).

Our response: Changed accordingly

-line 413: add "do" to read "and thus do not represent"

Our response: Changed accordingly

-Supplement page 8: can you please clarify how many SNPs were kept / discarded when you say "The analysis was restrained to SNPs lying on arctic fox scaffold 11 and that matched with a position on dog chromosome 5 during the BLAST".

Our response: 469 SNPs were located on scaffold 11 and BLASTed to dog chromosome 5. This information is now added, together with a reference to Table S14 were this number also is presented.

I am also not sure about the statement "SNPs on other scaffolds or that matched with different dog chromosomes would naturally appear as outliers in Figure S2" – first, do you rather mean Figure S4 (the GWAS) and second I am not sure 'outliers' is the right word here as SNPs within scaffold 11 that matched with another dog chromosome would not be highlighted in any way on this GWAS plot, and further, the significant SNPs in those other scaffolds aren't really outliers they are just not part of scaffold 11. Maybe rephrase?

Our response: Yes, the Figure reference was unfortunately not changed after adding new Figures to the Supplementary in the revision process. The correct reference is Figure S5 (the comparison between the arctic fox and dog genome). Our point here was that since we are comparing SNP positions on one dog chromosome vs one arctic fox scaffold, any SNPs on a different fox scaffold would logically not appear on the diagonal line. We have adjusted the text and hope that this and the correct Figure reference makes this understandable.

-Supplement page 9: can you please clarify that the reason your bar plots show 1125 individuals while your Table S3 shows 1181 is because you have 56 without phenotype? (maybe the scat samples?)

Our response: Yes, you are right. The difference comes from individuals for which we do not have the fur colour morph recorded (e.g. individuals only recorded by scat samples). An explanation has been added to Figure S6.

-page 14: Clarify which table you mean by Table SX?

Our response: The correct table reference is S6, we corrected this now. There were some additional mistakes in terms of referencing Figures and Tables in the Supplementary material. These are also corrected.

Very best wishes,
Anna Santure